# Robust Discriminative Representation Learning via Gradient Rescaling: An Emphasis Regularisation Perspective

## Abstract

It is fundamental and challenging to train robust and accurate Deep Neural Networks (DNNs) when semantically abnormal examples exist. Although great progress has been made, there is still one crucial research question which is not thoroughly explored yet: *What training examples should be focused on and how much more should they be emphasised to achieve robust learning?* In this work, we study this question and propose gradient rescaling (GR) to solve it. GR modifies the magnitude of logit vector's gradient to emphasise on relatively easier training data points when noise becomes more severe, which functions as explicit emphasis regularisation to improve the generalisation performance of DNNs. Apart from regularisation, we connect GR to examples weighting and designing robust loss functions. We empirically demonstrate that GR is highly anomaly-robust and outperforms the state-of-the-art by a large margin, e.g., increasing 7% on CIFAR-100 with 40% noisy labels. It is also significantly superior to standard regularisers in both clean and abnormal settings. Furthermore, we present comprehensive ablation studies to explore the behaviours of GR under different cases, which is informative for applying GR in real-world scenarios.

## 1 Introduction

DNNs have been successfully applied in diverse applications (Socher et al., 2011; Krizhevsky et al., 2012; LeCun et al., 2015). However, their success is heavily reliant on the quality of training data, especially accurate semantic labels for learning supervision. Unfortunately, on the one hand, maintaining the quality of semantic labels as the scale of training data increases is expensive and almost impossible when the scale becomes excessively large. On the other hand, it has been demonstrated that DNNs are capable of memorising the whole training data even when all training labels are random (Zhang et al., 2017). Therefore, DNNs struggle to discern meaningful data patterns and ignore semantically abnormal examples[1] simultaneously (Krueger et al., 2017; Arpit et al., 2017). Consequently, it becomes an inevitable demand for DNNs to hold robustness when training data contains anomalies (Larsen et al., 1998; Natarajan et al., 2013; Sukhbaatar & Fergus, 2014; Xiao et al., 2015; Patrini et al., 2017; Vahdat, 2017; Veit et al., 2017; Li et al., 2017).

Recently, great progress has been made towards robustness against anomalies when training DNNs (Krueger et al., 2017). There are three appealing perspectives in terms of their simplicity and effectiveness: 1) Examples weighting. For example, knowledge distilling from auxiliary models is popular for heuristically designing weighting schemes. However, it is challenging to select and train reliable auxiliary models in practice (Li et al., 2017; Malach & Shalev-Shwartz, 2017; Jiang et al., 2018; Ren et al., 2018; Han et al., 2018b). 2) Robust loss functions (Van Rooyen et al., 2015; Ghosh et al., 2017; Zhang & Sabuncu, 2018; Wang et al., 2019b); 3) Explicit regularisation techniques (Arpit et al., 2017; Zhang et al., 2018a). Although designing robust losses or explicit regularisation is easier and more flexible in practice, the performance is not the optimal yet.

---

[1]One training example is composed of an input and its corresponding label. A semantically abnormal example means the input is semantically unrelated to its label, which may come from corrupted input or label. For example, in Figure 3 in the supplementary material: 1) Out-of-distribution anomalies: An image may contain only background or an object which does not belong to any training class; 2) In-distribution anomalies: An image of class $a$ may be annotated to class $b$ or an image may contain more than one semantic object.

Regarding examples weighting, there is a core research question which is not well answered yet:

*What training examples should be focused on and how large the emphasis spread should be?*

In this work, we present a thorough study of this practical question under different settings. For better analysis, we propose two basic and necessary concepts: emphasis focus and spread with explicit definition in Sec. 3.2. They are conceptually introduced as follows:

**Emphasis focus.** It is a common practice to focus on harder instances when training DNNs (Shrivastava et al., 2016; Lin et al., 2017). When a dataset is clean, it achieves faster convergence and better performance to emphasise on harder examples because they own larger gradient magnitude, which means more information and a larger update step for model's parameters. However, when severe noise exists, as demonstrated in (Krueger et al., 2017; Arpit et al., 2017), DNNs learn simple meaningful patterns first before memorising abnormal ones. In other words, anomalies are harder to fit and own larger gradient magnitude in the later stage. Consequently, if we use the default sample weighting in categorical cross entropy (CCE) where harder samples obtain higher weights, anomalies tend to be fitted well especially when a network has large enough capacity. *That is why we need to move the emphasis focus towards relatively easier ones, which serves as emphasis regularisation.*

**Emphasis spread.** We term the weighting variance of training examples emphasis spread. The key concept is that we should not treat all examples equally, neither should we let only a few be emphasised and contribute to the training. *Therefore, when emphasis focus changes, the emphasis spread should be adjusted accordingly.*

We integrate emphasis focus and spread into a unified example weighting framework. Emphasis focus defines what training examples own higher weights while emphasis spread indicates how large variance over their weights. Specifically, we propose gradient rescaling (GR), which modifies the magnitude of logit vector's gradient. The logit vector is the output of the last fully connected (FC) layer of a network. *We remark that we do not design the weighting scheme heuristically from scratch. Instead, it is naturally motivated by the gradient analysis of several loss functions.*

Interestingly, GR can be naturally connected to examples weighting, robust losses, explicit regularisation: 1) The gradient magnitude of logit vector can be regarded as weight assignment that is built-in in loss functions (Gopal, 2016; Alain et al., 2016; Zhang et al., 2018b). Therefore, rescaling the gradient magnitude equals to adjusting the weights of examples; 2) A specific loss function owns a fixed gradient derivation. Adjusting the gradient can be treated as a more direct and flexible way of modifying optimisation objectives; 3) Instead of focusing on harder examples[2] by default, we can adjust emphasis focus to relative easier ones when noise is severe. GR serves as emphasis regularisation and is different from standard regularisers, e.g., L2 weight decay constraints on weight parameters and Dropout samples neural units randomly (Srivastava et al., 2014);

GR is simple yet effective. We demonstrate its effectiveness on diverse computer vision tasks using different net architectures: 1) Image classification with clean training data; 2) Image classification with synthetic symmetric label noise, which is more challenging than asymmetric noise evaluated by (Vahdat, 2017; Ma et al., 2018); 3) Image classification with real-world unknown anomalies, which may contain open-set noise (Wang et al., 2018), e.g., images with only background, or outliers, etc.; 4) Video person re-identification, a video retrieval task containing diverse anomalies. Beyond, we show that GR is notably better than other standard regularisers, e.g., L2 weight decay and dropout. Besides, to comprehensively understand GR's behaviours, we present extensive ablation studies.

**Main contribution.** Intuitively and principally, we claim that two basic factors, emphasis focus and spread, should be babysat simultaneously when it comes to examples weighting. To the best of our knowledge, we are the first to thoroughly study and analyse them together in a unified framework.

## 2 RELATED WORK

Aside from examples weighting, robust losses minimisation and explicit regularisation techniques, there are another two main perspectives for training robust and accurate DNNs when anomalies exist:

---

[2] An example's difficulty can be indicated by its loss (Shrivastava et al., 2016; Loshchilov & Hutter, 2016; Hinton, 2007), gradient magnitude (Gopal, 2016; Alain et al., 2016), or input-to-label relevance score (Lee et al., 2018). The input-to-label relevance score means the probability of an input belonging to its labelled class predicted by a current model. The difficulty of an example may change as the model learns. In summary, higher difficulty, larger loss, larger gradient magnitude, and lower input-to-label relevance score are equal concepts.

1) Robust training strategies (Miyato et al., 2018; Guo et al., 2018; Li et al., 2019; Thulasidasan et al., 2019); 2) Noise-aware modelling, and alternative label and parameter optimisation are popular when only label noise exists. Some methods focus on noise-aware modelling for correcting noisy labels or empirical losses (Larsen et al., 1998; Natarajan et al., 2013; Sukhbaatar & Fergus, 2014; Xiao et al., 2015; Vahdat, 2017; Veit et al., 2017; Goldberger & Ben-Reuven, 2017; Han et al., 2018a). However, it is non-trivial and time-consuming to learn a noise-aware model, which also requires prior extra information or some specific assumptions. For example, Masking (Han et al., 2018a) is assisted by human cognition to speculate the noise structure of noise-aware matrix while (Veit et al., 2017; Li et al., 2017; Lee et al., 2018; Hendrycks et al., 2018) exploit an extra clean dataset, which is a hyper-factor and hard to control in practice. Some other algorithms iteratively train the model and infer latent true labels (Wang et al., 2018; Tanaka et al., 2018). Those methods have made great progress on label noise. But they are not directly applicable to unknown diverse semantic anomalies, which covers both out-of-distribution and in-distribution cases.

## 2.1 REMARKS ON ROBUSTNESS THEOREMS CONDITIONED ON SYMMETRIC LOSSES

We note that (Ghosh et al., 2017) proposed some theorems showing that empirical risk minimization is robust when the loss function is symmetric and the noise type is label noise. However, they are not applicable for deep learning under arbitrary unknown noise: 1) We remark that we target at the problem of diverse or arbitrary abnormal examples, where an input may be out-of-distribution, i.e., not belonging to any training class. As a result, the symmetric losses custom-designed for label noise are not applicable. 2) GR is independent of empirical loss expressions as presented in Table 1. Therefore, one specific loss is merely an indicator of how far we are away from a specific minimisation objective. It has no impact on the robustness of the learning process since it has no direct influence on the gradient back-propagation. Similar to the prior work of rethinking generalisation (Zhang et al., 2017), we need to rethink robust training under diverse anomalies, where the robustness theorems conditioned on symmetric losses and label noise are not directly applicable.

## 3 EMPHASIS REGULARISATION BY GRADIENT RESCALING

**Notation**. We are given $N$ training examples $\mathbf{X} = \{(\mathbf{x}_i, y_i)\}_{i=1}^N$, where $(\mathbf{x}_i, y_i)$ denotes $i-$th sample with input $\mathbf{x}_i \in \mathbb{R}^D$ and label $y_i \in \{1, 2, ..., C\}$. $C$ is the number of classes. Let's consider a deep neural network $z$ composed of an embedding network $f(\cdot) : \mathbb{R}^D \to \mathbb{R}^K$ and a linear classifier $g(\cdot) : \mathbb{R}^K \to \mathbb{R}^C$, i.e., $\mathbf{z}_i = z(\mathbf{x}_i) = g(f(\mathbf{x}_i)) : \mathbb{R}^D \to \mathbb{R}^C$. Generally, the linear classifier is the last FC layer which produces the final output of $z$, i.e., logit vector $\mathbf{z} \in \mathbb{R}^C$. To obtain probabilities of a sample belonging to different classes, logit vector is normalised by a softmax function:

$$p(j|\mathbf{x}_i) = \exp(\mathbf{z}_{ij}) / \sum\nolimits_{m=1}^{C} \exp(\mathbf{z}_{im}). \tag{1}$$

$p(j|\mathbf{x}_i)$ is the probability of $\mathbf{x}_i$ belonging to class $j$. A sample's input-to-label relevance score is defined by $p_i = p(y_i|\mathbf{x}_i)$. In what follows, we will uncover the sample weighting in popular losses: CCE, Mean Absolute Error (MAE) and Generalised Cross Entropy (GCE) (Zhang & Sabuncu, 2018).

### 3.1 ANALYSING INTRINSIC SAMPLE WEIGHTING IN CCE, MAE AND GCE

**CCE**. The CCE loss with respect to $(\mathbf{x}_i, y_i)$, and its gradient with respect to $\mathbf{z}_{ij}$ are defined as:

$$L_{\mathrm{CCE}}(\mathbf{x}_i, y_i) = -\log p(y_i|\mathbf{x}_i) \quad \text{and} \quad \frac{\partial L_{\mathrm{CCE}}}{\partial \mathbf{z}_{ij}} = \begin{cases} p(y_i|\mathbf{x}_i) - 1, & j = y_i \\ p(j|\mathbf{x}_i), & j \neq y_i \end{cases}. \tag{2}$$

Therefore, we have $||\frac{\partial L_{\mathrm{CCE}}}{\partial \mathbf{z}_i}||_1 = 2(1 - p(y_i|\mathbf{x}_i)) = 2(1 - p_i)$. Here we choose L1 norm to measure the magnitude of gradient because of its simpler statistics and computation.

Since we back-propagate $\partial L_{\mathrm{CCE}}/\mathbf{z}_i$ to update the model's parameters, an example's gradient magnitude determines how much impact it has, i.e., its weight $w_i^{\mathrm{CCE}} = ||\frac{\partial L_{\mathrm{CCE}}}{\partial \mathbf{z}_i}||_1 = 2(1 - p_i)$. *In CCE, more difficult examples with smaller $p_i$ get higher weight.*

**MAE**. When it comes to MAE, the loss of $(\mathbf{x}_i, y_i)$ and gradient with respect to $\mathbf{z}_{im}$ are:

Table 1: Comparison between GR and other learning supervisions. 0∼0.5 and 0∼1 indicate the emphasis focus is adjustable and ranges from 0 to 0.5 and 0 to 1, respectively. Note that GR manipulates the gradients and is independent of specific losses, e.g., CCE, MAE and GCE.

| Supervision | Empirical loss | Gradient rescaling | Emphasis focus | Adjustable emphasis spread |
|---|---|---|---|---|
| CCE | CCE | No | 0 | No |
| MAE | MAE | No | 0.5 | No |
| GCE | GCE | No | 0∼0.5 | No |
| GR | CCE/MAE/GCE | Yes | 0∼1 | Yes |

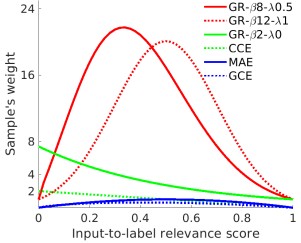
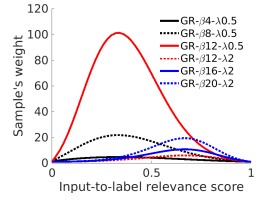
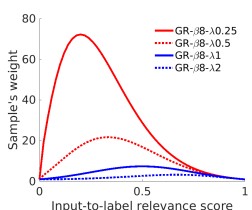

(a) GR, CCE, MAE, GCE. We show 3 settings of GR: $(\beta = 2, \lambda = 0)$, $(\beta = 8, \lambda = 0.5)$ and $(\beta = 12, \lambda = 1)$. Their corresponding emphasis focuses are 0, 0∼0.5 and 0.5.

(b) GR when fixing $\lambda = 0.5$ (emphasis focus is within 0∼0.5) or $\lambda = 2$ (emphasis focus is within 0.5∼1).

(c) GR when fixing $\beta = 8$. When $\lambda$ increases, the emphasis focus moves towards 1 and emphasis spread drops.

Figure 1: A sample's weight $w_i$ along with its input-to-label relevance score $p_i$. GR is a unified sample reweighting framework from the perspective of gradient rescaling, where the emphasis focus and spread can be adjusted by choosing proper $\lambda$ and $\beta$ in practice. Better viewed in colour.

$$L_{\text{MAE}}(\mathbf{x}_i, y_i) = 2(1 - p(y_i|\mathbf{x}_i)) \quad \text{and} \quad \frac{\partial L_{\text{MAE}}}{\partial \mathbf{z}_{ij}} = \begin{cases} 2p(y_i|\mathbf{x}_i)(p(y_i|\mathbf{x}_i) - 1), & j = y_i \\ 2p(y_i|\mathbf{x}_i)p(j|\mathbf{x}_i), & j \neq y_i \end{cases}. \quad (3)$$

Therefore, $w_i^{\text{MAE}} = ||\frac{\partial L_{\text{MAE}}}{\partial \mathbf{z}_i}||_1 = 4p(y_i|\mathbf{x}_i)(1 - p(y_i|\mathbf{x}_i)) = 4p_i(1 - p_i)$. *In MAE, those images whose input-to-label relevance scores are 0.5 become the emphasis focus.*

**GCE**. In GCE, the loss calculation of $(\mathbf{x}_i, y_i)$ and gradient with respect to logit vector $\mathbf{z}_i$ are:

$$L_{\text{GCE}}(\mathbf{x}_i, y_i) = \frac{1 - p(y_i|\mathbf{x}_i)^q}{q} \quad \text{and} \quad \frac{\partial L_{\text{GCE}}}{\partial \mathbf{z}_{ij}} = \begin{cases} p(y_i|\mathbf{x}_i)^q(p(y_i|\mathbf{x}_i) - 1), & j = y_i \\ p(y_i|\mathbf{x}_i)^q p(j|\mathbf{x}_i), & j \neq y_i \end{cases}, \quad (4)$$

where $q \in [0, 1]$. Therefore, $w_i^{\text{GCE}} = ||\frac{\partial L_{\text{GCE}}}{\partial \mathbf{z}_i}||_1 = 2p(y_i|\mathbf{x}_i)^q(1 - p(y_i|\mathbf{x}_i)) = 2p_i^q(1 - p_i)$. In this case, the emphasis focus can be adjusted from 0 to 0.5 when $q$ ranges from 0 to 1. However, in their practice (Zhang & Sabuncu, 2018), instead of using this naive version, a truncated one is applied:

$$L_{\text{GCE}_{\text{trunc}}}(\mathbf{x}_i, y_i) = \begin{cases} L_q(p_i), & p_i > 0.5 \\ L_q(0.5), & p_i \leq 0.5 \end{cases} \quad \text{and} \quad L_q(\gamma) = (1 - \gamma^q)/q, \quad (5)$$

The loss of an example with $p_i \leq 0.5$ is constant so that its gradient is zero, which means it is dropped and does not contribute to the training. The main drawback is that at the initial stage, the model is not well learned so that the predicted $p_i$ of most samples are smaller than $0.5$. *To address it, alternative convex search is exploited for iterative data pruning and parameters optimisation, making it quite complex and less appealing in practice.*

The derivation details of Eq. (2), (3), (4) are presented in Section B of the supplementary material.

## 3.2 GRADIENT RESCALING FOR EMPHASIS REGULARISATION

A loss function provides supervision information by its derivative with respect to a network's output. Therefore, there are two perspectives for improving the supervision information: 1) Modifying the loss format to improve its corresponding derivative; 2) Manipulating the gradient straightforwardly. In this work, we choose to control the gradient, which is more direct and flexible.

According to Eq. (2), (3), (4), the gradients of CCE, MAE and GCE share the same direction. Our proposal GR unifies them from the gradient perspective. Being independent of loss formulas, a sample's gradient is rescaled linearly so that its weight is $w_i^{\text{GR}}$:

$$w_i^{\text{GR}} = g(\beta p_i^\lambda (1 - p_i)) => \frac{\partial L}{\partial \mathbf{z}_i} = \frac{\partial L_{\text{CCE}}}{\partial \mathbf{z}_i} \frac{w_i^{\text{GR}}}{w_i^{\text{CCE}}} = \frac{\partial L_{\text{MAE}}}{\partial \mathbf{z}_i} \frac{w_i^{\text{GR}}}{w_i^{\text{MAE}}} = \frac{\partial L_{\text{GCE}}}{\partial \mathbf{z}_i} \frac{w_i^{\text{GR}}}{w_i^{\text{GCE}}}, \quad (6)$$

where $\lambda, \beta$ are hyper-parameters for controlling the emphasis focus and spread, respectively. By choosing a larger $\lambda$ when more anomalies exist, GR regularises examples weighting by moving emphasis focus toward relatively easier training data points, thus embracing noise-robustness.

For clarification, we explicitly define the emphasis focus and spread over training examples:

**Definition 1** (Emphasis Focus $\psi$). *The emphasis focus refers to those examples that own the largest weight. Since an example's weight is determined by its input-to-label relevance score $p_i$, for simplicity, we define the emphasis focus to be an input-to-label score to which the largest weight is assigned, i.e., $\psi = \arg\max_{p_i} w_i^{\text{GR}} \in [0, 1)$.*

**Definition 2** (Emphasis Spread $\sigma$). *The emphasis spread is the weight variance over all training instances in a mini-batch, i.e., $\sigma = \text{E}((w_i^{\text{GR}} - \text{E}(w_i^{\text{GR}}))^2)$, where $\text{E}(\cdot)$ denotes the expectation value of a variable.*

With these definitions, we differentiate GR with other methods in Table 1. We show the sample weighting curves of GR with different settings in Figure 1. As shown in Figure 1c, the emphasis spread declines as $\lambda$ increases. Therefore, we choose larger $\beta$ values when $\lambda$ is larger in Sec. 4.2.1. Principally, transformation $g$ could be designed as any monotonically increasing function. Because the non-linear exponential mapping can change the overall weights' variance and relative weights between any two examples, we choose $g(\cdot) = \exp(\cdot)$, which works well in our practice. By integral, the exact loss format is an error function (non-elementary). We summarise several existing cases as follows (the ellipsis refers to other potential options which can be explored in the future):

$$w_i^{\text{GR}} = \begin{cases} w_i^{\text{CCE}}, & \beta = 2, \lambda = 0, g = \text{identity} \\ w_i^{\text{MAE}}, & \beta = 4, \lambda = 1, g = \text{identity} \\ w_i^{\text{GCE}}, & \beta = 1, 1 \geq \lambda \geq 0, g = \text{identity} \\ \exp(\beta p_i^\lambda (1 - p_i)), & \beta \geq 0, \lambda \geq 0, g = \exp \\ ... \end{cases} \quad (7)$$

### 3.3 Why Does GR Contribute to Robust Learning?

Let's regard a deep network $z$ as a black box, which produces $C$ logits. $C$ is the class number. Then during gradient back-propagation, an example's impact on the update of $z$ is determined by its gradient w.r.t. the logit vector. The impact can be decomposed into two factors, i.e., gradient direction and magnitude. To reduce the impact of a noisy sample, we can either reduce its gradient magnitude or amend its gradient direction. In this work, *inspired by the analysis of CCE, MAE and GCE, which only differ in the gradient magnitude while perform quite differently, leading to a natural motivation that gradient magnitude matters. That is why we explore rescaling the gradient magnitude as illustrated in Figure 1.* It is worth studying amending gradient directions in the future.

## 4 Experiments

### 4.1 Image Classification with Clean Training Data

**Datasets.** We test on CIFAR-10 and CIFAR-100 (Krizhevsky, 2009), which contain 10 and 100 classes, respectively. In CIFAR-10, the training data contains 5k images per class while the test set includes 1k images per class. In CIFAR-100, there are 500 images per class for training and 100 images per class for testing.

**Implementation details.** On CIFAR-10, following (He et al., 2016), we adopt ResNet-20 and ResNet-56 as backbones so that we can compare fairly with their reported results. On CIFAR-100, we follow D2L (Ma et al., 2018) to choose ResNet-44 and compare with its reported results. We also use an SGD optimiser with momentum 0.9 and weight decay $10^{-4}$. The learning rate is initialised with 0.1, and multiplied with 0.1 every 5k iterations. We apply the standard data augmentation as in (He et al., 2016; Ma et al., 2018): The original images are padded with 4 pixels on every side, followed by a random crop of $32 \times 32$ and horizontal flip. The batch size is 128.

Table 2: Classification accuracies (%) of CCE, and GR on clean CIFAR-10 and CIFAR-100. $\lambda = 0$ means the emphasis focus is 0 where we fix $\beta = 2$. $\beta = 0$ means all examples are treated equally.

| Dataset | Backbone | CCE | GR ($\lambda = 0$) | GR ($\beta = 0$) |
|---------|----------|-----|--------------------|------------------|
| CIFAR-10 | ResNet-20 | 91.8 | 91.8 | 91.0 |
| | ResNet-56 | 92.4 | 92.5 | 91.9 |
| CIFAR-100 | ResNet-44 | 68.1 | 68.4 | 66.4 |

**Results.** Our purpose is to show GR can achieve competitive performance with CCE under clean data to demonstrate its general applicability. As reported in D2L, all noise-tolerant proposals (Patrini et al., 2017; Reed et al., 2015; Ma et al., 2018) perform similarly with CCE when training labels are clean. Therefore we do not present other related competitors here. Our reimplemented results are shown in Table 2. For reference, the reported results in (He et al., 2016) on CIFAR-10 with CCE are 91.3% for ResNet-20 and 93.0% for ResNet-56. In D2L, the result on CIFAR-100 with ResNet-44 is 68.2%. Our reimplemented performance of CCE is only slightly different. For GR, we observe the best performance when emphasis focus is 0, i.e., $\lambda = 0$. Furthermore, it is insensitive to a wide range of emphasis spreads according to our observations in Figure 5 in the supplementary material.

*Treating training examples equally.* As shown in Table 2, we obtain competitive performance by treating all training examples equally when $\beta = 0$. This is quite interesting and motivates us that sample differentiation and reweighting work much better only when noise exists.

### 4.2 IMAGE CLASSIFICATION WITH SYNTHETIC SYMMETRIC LABEL NOISE

**Symmetric noise generation.** Given a probability $r$, the original label of an image is changed to one of the other class labels uniformly following (Tanaka et al., 2018; Ma et al., 2018). $r$ denotes the noise rate. Symmetric label noise generally exists in large-scale real-world applications where the dataset scale is so large that label quality is hard to guarantee. It is also demonstrated in (Vahdat, 2017) that it is more challenging than asymmetric noisy labels (Reed et al., 2015; Patrini et al., 2017), which assume that label errors only exist within a predefined set of similar classes. All augmented training examples share the same label as the original one.

#### 4.2.1 EMPIRICAL ANALYSIS OF GR ON CIFAR-10

To understand GR well empirically, we explore the behaviours of GR on CIFAR-10 with $r = 20\%, 40\%, 60\%, 80\%$, respectively. We use ResNet-56 which has larger capacity than ResNet-20.

**Design choices.** We mainly analyse the impact of different emphasis focuses for different noise rates. We explore 5 emphasis focuses by setting $\beta = 0$ or different $\lambda$: 1) None: $\beta = 0$. There is no emphasis focus since all examples are treated equally; 2) 0: $\lambda = 0$; 3) 0~0.5: $\lambda = 0.5$; 4) 0.5: $\lambda = 1$; 5) 0.5~1: $\lambda = 2$. We remark that when $\lambda$ is larger, the emphasis focus is higher, leading to relatively easier training data points are emphasised. As shown in Figure 1, *when emphasis focus changes, emphasis spread changes accordingly*. Therefore, to set a proper spread for each emphasis focus, we try 4 emphasis spread and choose the best one[3] to compare the impact of emphasis focus.

**Results analysis.** We show the results in Table 3. The intact training set serves as a validation set and we observe that its accuracy is always consistent with the final test accuracy. This motivates us that we can choose our model's hyper-parameters $\beta, \lambda$ via a validation set in practice. We display the training dynamics in Figure 2. We summarise our observations as follows:

*Fitting and generalisation.* We observe that CCE always achieves the best accuracy on corrupted training sets, which indicates that CCE has a strong data fitting ability even if there is severe noise (Zhang et al., 2017). As a result, CCE has much worse final test accuracy than most models.

*Emphasising on harder examples.* When there exist abnormal training examples, we obtain the worst final test accuracy if emphasis focus is 0, i.e., CCE and GR with $\lambda = 0$. This unveils that in applications where we have to learn from noisy training data, it will hurt the model's generalisation dramatically if we use CCE or simply focus on harder training data points.

*Emphasis focus.* When noise rate is 0, 20%, 40%, 60%, and 80%, we obtain the best final test accuracy when $\lambda = 0$, $\lambda = 0.5$, $\lambda = 1$, $\lambda = 2$, and $\lambda = 2$, respectively. This demonstrates that

---

[3]Since there is a large interval between different $\beta$ in our four trials, we deduce that the chosen one is not the optimal. The focus of this work is not to optimize the hyper-parameters.

Table 3: Results of CCE, GR on CIFAR-10 with noisy labels. For every model, we show its best test accuracy during training and the final test accuracy when training terminates, which are indicated by 'Best' and 'Final', respectively. We also present the results on corrupted training sets and original intact one. The overlap rate between corrupted and intact sets is $(1 - r)$. Therefore, we can regard the intact training set as a validation set. When $\lambda$ is larger, $\beta$ should be larger as shown in Figure 1c.

| Noise Rate $r$ | Emphasis Focus | Model | Testing Accuracy (%) | | Accuracy on Training Sets (%) | |
|---|---|---|---|---|---|---|
| | | | Best | Final | Corrupted/Fitting | Intact/Validation |
| | 0 | CCE | 86.5 | 76.8 | **95.7** | 80.6 |
| 20% | None | GR ($\beta$=0) | 83.5 | 58.1 | 50.6 | 60.2 |
| | 0 ($\lambda = 0$) | GR ($\beta = 2$) | 84.9 | 76.4 | 85.3 | 80.5 |
| | 0~0.5 ($\lambda = 0.5$) | GR ($\beta = 12$) | **89.4** | **87.8** | 81.5 | **95.0** |
| | 0.5 ($\lambda = 1$) | GR ($\beta = 16$) | 87.3 | 86.7 | 78.4 | 93.8 |
| | 0.5~1 ($\lambda = 2$) | GR ($\beta = 24$) | 85.8 | 85.5 | 76.0 | 91.4 |
| | 0 | CCE | 82.8 | 60.9 | **83.0** | 64.4 |
| 40% | None | GR ($\beta$=0) | 71.8 | 44.9 | 31.3 | 45.8 |
| | 0 ($\lambda = 0$) | GR ($\beta = 1$) | 78.4 | 65.6 | 63.3 | 66.6 |
| | 0~0.5 ($\lambda = 0.5$) | GR ($\beta = 12$) | **85.1** | 79.9 | 67.7 | 85.7 |
| | 0.5 ($\lambda = 1$) | GR ($\beta = 16$) | 84.7 | **83.3** | 60.3 | **88.9** |
| | 0.5~1 ($\lambda = 2$) | GR ($\beta = 20$) | 52.7 | 52.7 | 35.4 | 53.6 |
| | 0 | CCE | 69.5 | 37.2 | **84.1** | 40.5 |
| 60% | None | GR ($\beta$=0) | 69.9 | 57.9 | 40.1 | 58.6 |
| | 0 ($\lambda = 0$) | GR ($\beta = 0.5$) | 72.3 | 53.9 | 42.1 | 55.1 |
| | 0~0.5 ($\lambda = 0.5$) | GR ($\beta = 12$) | 77.5 | 58.5 | 55.5 | 62.6 |
| | 0.5 ($\lambda = 1$) | GR ($\beta = 12$) | 71.9 | 70.0 | 41.0 | 73.9 |
| | 0.5~1 ($\lambda = 2$) | GR ($\beta = 12$) | **80.2** | **72.5** | 44.9 | **75.4** |
| | 0 | CCE | 36.1 | 16.1 | **54.3** | 18.4 |
| 80% | None | GR ($\beta$=0) | 44.4 | 28.2 | 20.6 | 28.8 |
| | 0 ($\lambda = 0$) | GR ($\beta = 0.5$) | 46.2 | 21.3 | 27.8 | 23.1 |
| | 0~0.5 ($\lambda = 0.5$) | GR ($\beta = 8$) | **51.6** | 22.4 | 46.1 | 24.4 |
| | 0.5 ($\lambda = 1$) | GR ($\beta = 8$) | 35.5 | 31.5 | 19.8 | 32.3 |
| | 0.5~1 ($\lambda = 2$) | GR ($\beta = 12$) | 33.0 | **32.8** | 14.2 | **32.6** |

when noise rate is higher, we can improve a model's robustness by moving emphasis focus towards relatively less difficult examples with a larger $\lambda$, which is informative in practice.

*Emphasis spread.* As displayed in Table 3 and Figures 7-10 in the supplementary material, emphasis spread also matters a lot when fixing emphasis focus, i.e., fixing $\lambda$. For example in Table 3 , when $\lambda = 0$, although focusing on harder examples similarly with CCE, GR can outperform CCE by modifying the emphasis spread. As shown in Figures 7-10, some models even collapse and cannot converge if the emphasis spread is not rational.

### 4.2.2 COMPETING WITH THE STATE-OF-THE-ART ON CIFAR-10

**Implementation details.** We follow the same settings as MentorNet (Jiang et al., 2018) to compare fairly with its reported results. Optimiser and data augmentation are described in Section 4.1.

**Competitors.** FullModel is the standard CCE trained using L2 weight decay and dropout (Srivastava et al., 2014). Forgetting (Arpit et al., 2017) searches the dropout parameter in the range of (0.2-0.9). Self-paced (Kumar et al., 2010), Focal Loss (Lin et al., 2017), and MentorNet (Jiang et al., 2018) are representatives of example reweighting algorithms. Reed Soft (Reed et al., 2015) is a weakly-supervised learning method. All methods use GoogLeNet V1 (Szegedy et al., 2015).

**Results.** We compare the results under different noise rates in Table 4. GR with fixed hyper-parameters $\beta = 8$, $\lambda = 0.5$ outperforms the state-of-the-art GCE by a large margin, especially when label noise becomes severe. Better results can be expected when optimising the hyper-parameters for each case. We remark that FullModel (naive CCE) (Jiang et al., 2018) was trained with L2 weight decay and dropout. However, GR's regularization effect is much better in both clean and noisy cases.

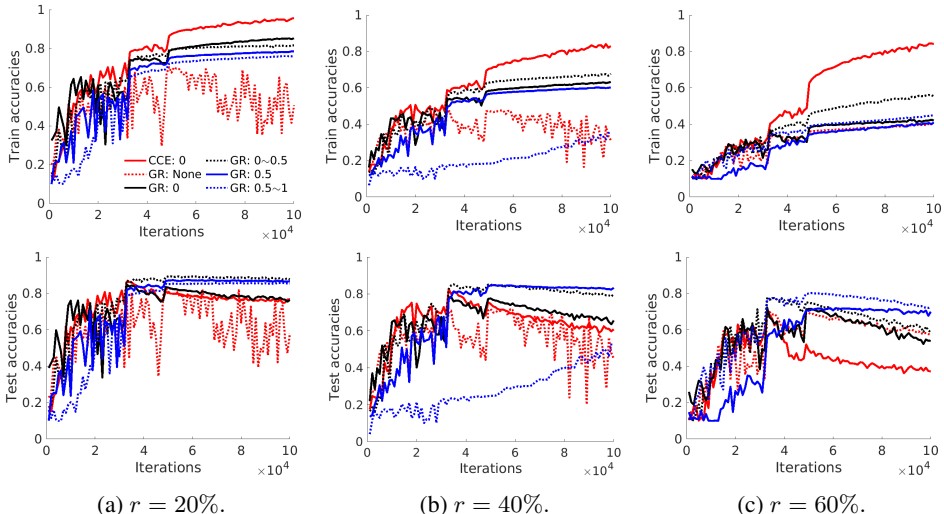

Figure 2: The learning dynamics of ResNet-56 on CIFAR-10, i.e., training and testing accuracies along with training iterations. The legend in the top left is shared by all subfigures. 'xxx: yyy' means 'method: emphasis focus'. The results of $r = 80\%$ are shown in Figure 6 in the supplementary material. We have two key observations: 1) When noise rate increases, better generalisation is obtained with higher emphasis focus, i.e., focusing on relatively easier examples; 2) Both overfitting and underfitting lead to bad generalisation. For example, 'CCE: 0' fits training data much better than the others while 'GR: None' generally fits it unstably or a lot worse. Better viewed in colour.

### 4.2.3 COMPETING WITH THE STATE-OF-THE-ART ON CIFAR-100

**Implementation details.** Most baselines have been reimplemented in (Ma et al., 2018) with the same settings. Therefore, for direct comparison, we follow exactly their experimental configurations and use ResNet-44 (He et al., 2016). Optimiser and data augmentation are described in Section 4.1. We repeat training and evaluation 5 times where different random seeds are used for generating noisy labels and model's initialisation. The mean test accuracy and standard deviation are reported.

**Competitors.** We compare with D2L (Ma et al., 2018), GCE (Zhang & Sabuncu, 2018), and other baselines reimplemented in D2L: 1) Standard CCE (Ma et al., 2018); 2) Forward (Patrini et al., 2017) uses a noise-transition matrix to multiply the network's predictions for label correction; 3) Backward (Patrini et al., 2017) applies the noise-transition matrix to multiply the CCE losses for loss correction; 4) Bootstrapping (Reed et al., 2015) trains models with new labels generated by a convex combination of the original ones and their predictions. The convex combination can be soft (Boot-soft) or hard (Boot-hard); 5) D2L (Ma et al., 2018) achieves noise-robustness from a novel perspective of restricting the dimensionality expansion of learned subspaces during training and is the state-of-the-art; 6) Since GCE outperforms MAE (Zhang & Sabuncu, 2018), we only reimplement GCE for comparison; 7) SL (Wang et al., 2019c) boosts CCE symmetrically with a noise-robust counterpart, i.e., reverse cross entropy.

**Results.** We compare the results of GR and other algorithms in Table 5. GR outperforms other competitors by a large margin, especially when label noise is severe, e.g., $r = 40\%$ and $60\%$. More importantly, we highlight that *GR is much simpler without any extra information*. Compared with Forward and Backward, GR does not need any prior knowledge about the noise-transition matrix. Bootstrapping targets at label correction and is time-consuming. D2L estimates the local intrinsic dimensionality every $b$ mini-batches and checks the turning point for dimensionality expansion every $e$ epochs. However, $b$ and $e$ are difficult to choose and iterative monitoring is time-consuming.

### 4.3 IMAGE CLASSIFICATION WITH REAL-WORLD UNKNOWN NOISE

**Dataset.** Clothing 1M (Xiao et al., 2015) contains 1 million images. It is an industrial-level dataset and its noise structure is agnostic. According to (Xiao et al., 2015), around 61.54% training labels are reliable, i.e., the noise rate is about 38.46%. There are 14 classes from several online shopping websites. In addition, there are 50k, 14k, and 10k images with clean labels for training, validation,

Table 4: The results of GR and other noise-robust approaches on CIFAR-10 using GoogLeNet V1.

| Noise rate $r$ | FullModel (naive CCE) | Forgetting | Self-paced | Focal Loss | Reed Soft | MentorNet PD | Mentor DD | GCE | GR ($\beta = 8, \lambda = 0.5$) |
|---|---|---|---|---|---|---|---|---|---|
| 0 | 0.81 | – | – | – | – | – | – | 0.83 | **0.85** |
| 20% | 0.76 | 0.76 | 0.80 | 0.77 | 0.78 | 0.79 | 0.79 | 0.81 | **0.83** |
| 40% | 0.73 | 0.71 | 0.74 | 0.74 | 0.73 | 0.74 | 0.76 | 0.78 | **0.79** |
| 80% | 0.42 | 0.44 | 0.33 | 0.40 | 0.39 | 0.44 | 0.46 | 0.50 | **0.57** |

Table 5: The accuracies (%) of GR and recent approaches on CIFAR-100. The results of fixed parameters ($\beta = 8, \lambda = 0.5$) are shown in the second last column. With a little effort for optimising $\beta$ and $\lambda$, the results and corresponding parameters are presented in the last column. The trend is consistent with Table 3: When $r$ raises, we can increase $\beta, \lambda$ for better robustness. The increasing scale is much smaller. This is because CIFAR-100 has 100 classes so that its distribution of $p_i$ (input-to-label relevance score) is different from CIFAR-10 after softmax normalisation.

| Noise rate $r$ | CCE | GCE | Forward | Backward | Boot-hard | Boot-soft | D2L | SL | GR ($\beta = 8, \lambda = 0.5$) | GR ($\beta, \lambda$) |
|---|---|---|---|---|---|---|---|---|---|---|
| 20% | 52.9±0.2 | 53.4±0.3 | 60.3±0.2 | 58.7±0.3 | 58.5±0.4 | 57.3±0.3 | 62.2±0.4 | 60.0±0.2 | 62.6±0.3 | **64.1**±0.2 (6, 0.3) |
| 40% | 42.9±0.2 | 47.0±0.2 | 51.3±0.3 | 45.4±0.2 | 44.4±0.1 | 41.9±0.1 | 52.0±0.3 | 53.7±0.1 | 59.3±0.2 | **60.0**±0.1 (6, 0.4) |
| 60% | 30.1±0.2 | 41.0±0.2 | 41.2±0.3 | 34.5±0.2 | 36.7±0.3 | 32.3±0.1 | 42.3±0.2 | 41.5±0.0 | **49.9**±0.3 | **49.9**±0.3 (8, 0.5) |

and testing, respectively. Here, we follow and compare with existing methods that only learn from noisy training data since we would like to avoid exploiting auxiliary information.

**Implementation details.** We train ResNet-50 (He et al., 2016) and follow exactly the same settings as (Patrini et al., 2017; Tanaka et al., 2018): 1) Initialisation: ResNet-50 is initialised by publicly available model pretrained on ImageNet (Russakovsky et al., 2015); 2) Optimisation: A SGD optimiser with a momentum of 0.9 and a weight decay of $10^{-3}$ is applied. The learning rate starts at $10^{-3}$ and is divided by 10 after 5 epochs. Training terminates at 10 epochs; 3) Standard data augmentation: We first resize a raw input image to $256 \times 256$, and then crop it randomly at $224 \times 224$ followed by random horizontal flipping. The batch size is 64 due to memory limitation. *Since the noise rate is around 38.46%, we simply set $\lambda = 1, \beta = 16$ following Table 3 when noise rate is 40%.*

**Competitors.** We compare with other noise-robust algorithms that have been evaluated on Clothing 1M with similar settings: 1) Standard CCE (Patrini et al., 2017); 2) Since Forward outperforms Backward on Clothing 1M (Patrini et al., 2017), we only present the result of Forward; 3) S-adaptation applies an additional softmax layer to estimate the noise-transition matrix (Goldberger & Ben-Reuven, 2017); 4) Masking is a human-assisted approach that conveys human cognition to speculate the structure of the noise-transition matrix (Han et al., 2018a). 5) Label optimisation (Tanaka et al., 2018) learns latent true labels and model's parameters iteratively. Two regularisation terms are added for label optimisation and adjusted in practice.

**Results.** The results are compared in Table 6. Under real-world agnostic noise, GR also outperforms the state-of-the-art. It is worth mentioning that the burden of noise-transition matrix estimation in Forward and S-adaptation is heavy due to alternative optimisation steps, and such estimation is non-trivial without big enough data. Masking exploits human cognition of a structure prior and reduces the burden of estimation, nonetheless its performance is not competitive. Similarly, Label Optimisation requires alternative optimisation steps and is time-consuming.

### 4.4 VIDEO RETRIEVAL WITH DIVERSE ANOMALIES

**Dataset and evaluation settings.** MARS contains 20,715 videos of 1,261 persons (Zheng et al., 2016). There are 1,067,516 frames in total. Because person videos are collected by tracking and detection algorithms, abnormal examples exist as shown in Figure 3 in the supplementary material. We remark that there are some anomalies containing only background or an out-of-distribution person. Exact noise type and rate are unknown. Following standard settings, we use 8,298 videos of 625 persons for training and 12,180 videos of the other 636 persons for testing. We report the cumulated matching characteristics (CMC) and mean average precision (mAP) results.

Table 6: The classification accuracy (%) on Clothing1M with ResNet-50. CCE and GCE were reported in (Patrini et al., 2017) and (Wang et al., 2019c), respectively. CCE* and GCE* are our reproduced results using the Caffe framework (Jia et al., 2014).

| Boot -soft | Forward | Bilevel Optimisation | S-adaptation | Masking | Joint Optimisation | CCE | CCE* | GCE | GCE* | SL | GR |
|---|---|---|---|---|---|---|---|---|---|---|---|
| 69.1 | 69.8 | 69.9 | 70.4 | 71.1 | 72.2 | 68.9 | 71.7 | 69.8 | 72.5 | 71.0 | **73.2** |

Table 7: The video retrieval results on MARS. For fair comparison, all other methods use GoogLeNet V2 except DRSA and CAE using more complex ResNet-50.

| Metric | CCE | MAE | GCE | DRSA | CAE | OSM+CAA | GR |
|---|---|---|---|---|---|---|---|
| mAP (%) | 58.1 | 12.0 | 31.6 | 65.8 | 67.5 | 72.4 | **72.8** |
| CMC-1 (%) | 73.8 | 26.0 | 51.5 | 82.3 | 82.4 | **84.7** | 84.3 |

**Implementation details.** Following (Liu et al., 2017; Wang et al., 2019a), we train GoogLeNet V2 (Ioffe & Szegedy, 2015) and treat a video as an image set, which means we use only appearance information without exploiting latent temporal information. A video's representation is simply the average fusion of its frames' representations. The learning rate starts from 0.01 and is divided by 2 every 10k iterations. We stop training at 50k iterations. We apply an SGD optimiser with a weight decay of 0.0005 and a momentum of 0.9. The batch size is 180. We use standard data augmentation: a $227 \times 227$ crop is randomly sampled and flipped after resizing an original image to $256 \times 256$. Training settings are the same for each method. We implement GCE with its reported best settings. At testing, following (Wang et al., 2019a; Movshovitz-Attias et al., 2017; Law et al., 2017), we first $L_2$ normalise videos' features and then calculate the cosine similarity between every two of them.

**Results.** The results are displayed in Table 7. Although DRSA (Li et al., 2018) and CAE (Chen et al., 2018) exploit extra temporal information by incorporating attention mechanisms, GR is superior to them in terms of both effectiveness and simplicity. OSM+CAA (Wang et al., 2019a) is the only comparable method. However, OSM+CAA combines CCE and weighted contrastive loss to address anomalies, thus being more complex than GR. In addition, we highlight that one query may have multiple matching instances in the MARS benchmark. Consequently, mAP is a more reliable and accurate performance assessment. GR is the best in terms of mAP.

### 4.5 BEATING STANDARD REGULARISERS UNDER LABEL NOISE

In Table 8, we compare our proposed regulariser GR with other standard ones, i.e., L2 weight decay and Dropout (Srivastava et al., 2014). We set the dropout rate to 0.2 and L2 weight decay rate to $10^{-4}$. For GR, as mentioned in Section 4.2.3, we fix $\beta = 8, \lambda = 0.5$. Interestingly, Dropout+L2 achieves 52.8% accuracy, which is even better than the state-of-the-art in Table 5, i.e., D2L with 52.0% accuracy. However, GR is better than those standard regularisers and their combinations significantly. GR works best when it is together with L2 weight decay.

Table 8: Results of GR and other standard regularisers on CIFAR-100. We set $r = 40\%$, i.e., the label noise is severe but not belongs to the majority. We train ResNet-44. We report the average test accuracy and standard deviation (%) over 5 trials. Baseline means CCE without regularisation.

| Baseline | L2 | Dropout | Dropout+L2 | GR | GR+L2 | GR+Dropout | GR+L2+Dropout |
|---|---|---|---|---|---|---|---|
| 44.7±0.1 | 51.5±0.4 | 46.7±0.5 | 52.8±0.4 | 55.7±0.3 | **59.3**±0.2 | 54.3±0.4 | 58.3±0.3 |

## 5 CONCLUSION

In this work, we present three main contributions: 1) We analyse and answer a core research question: What training examples should be focused on and how large the emphasis spread should be? 2) We uncover and analyse that two basic factors, emphasis focus and spread, should be babysat simultaneously when it comes to examples weighting. Consequently, we propose a simple yet effective gradient rescaling framework serving as emphasis regularisation. 3) Extensive experiments on different tasks using different network architectures are reported for better understanding and demonstration of GR's effectiveness, which are also valuable for applying GR in practice.

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

# Supplementary Material for
# Robust Discriminative Representations Learning via Gradient Rescaling: An Emphasis Regularisation Perspective

## A   DISPLAY OF SEMANTICALLY ABNORMAL TRAINING EXAMPLES

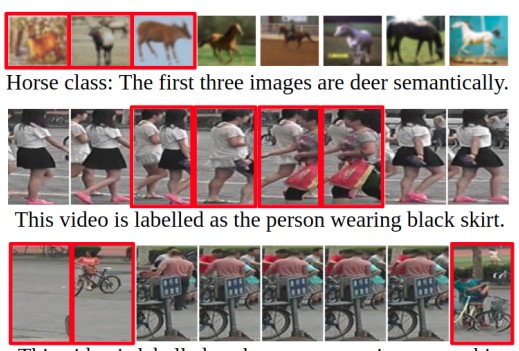

Horse class: The first three images are deer semantically.

This video is labelled as the person wearing black skirt.

This video is labelled as the person wearing green shirt.

Figure 3: Diverse semantically abnormal training examples highlighted by red boxes. The 1st row shows synthetic abnormal examples from corrupted CIFAR-10 (Krizhevsky, 2009). The 2nd and 3rd rows present realistic abnormal examples from video person re-identification benchmark MARS (Zheng et al., 2016).
***Out-of-distribution anomalies***: 1) The first image in the 3rd row contains only background and no semantic information at all. 2) The 2nd first image or the last one in the 3rd row may contain a person that does not belong to any person in the training set.
***In-distribution anomalies***: 1) Some images of deer class are wrongly annotated to horse class. 2) We cannot decide the object of interest without any prior when an image contains more than one object, e.g., some images contain two persons in the 2nd row.

## B   DERIVATION DETAILS OF SOFTMAX, CCE, MAE AND GCE

### B.1   DERIVATION OF SOFTMAX NORMALISATION

Based on Eq. (1), we have

$$p(y_i|\mathbf{x}_i)^{-1} = 1 + \sum_{j \neq y_i} \exp(\mathbf{z}_{ij} - \mathbf{z}_{iy_i}). \tag{8}$$

For left and right sides of Eq. (8), we calculate their derivatives w.r.t. $\mathbf{z}_{ij}$ simultaneously.

If $j = y_i$,

$$\frac{-1}{p(y_i|\mathbf{x}_i)^2} \frac{\partial p(y_i|\mathbf{x}_i)}{\mathbf{z}_{iy_i}} = -\sum_{j \neq y_i} \exp(\mathbf{z}_{ij} - \mathbf{z}_{iy_i})$$

$$=> \frac{\partial p(y_i|\mathbf{x}_i)}{\mathbf{z}_{iy_i}} = p(y_i|\mathbf{x}_i)(1 - p(y_i|\mathbf{x}_i)). \tag{9}$$

If $j \neq y_i$,

$$\frac{-1}{p(y_i|\mathbf{x}_i)^2} \frac{\partial p(y_i|\mathbf{x}_i)}{\mathbf{z}_{ij}} = \exp(\mathbf{z}_{ij} - \mathbf{z}_{iy_i})$$

$$=> \frac{\partial p(y_i|\mathbf{x}_i)}{\mathbf{z}_{ij}} = -p(y_i|\mathbf{x}_i)p(j|\mathbf{x}_i). \tag{10}$$

*In summary*, the derivation of softmax layer is:

$$\frac{\partial p(y_i|\mathbf{x}_i)}{\partial \mathbf{z}_{ij}} = \begin{cases} p(y_i|\mathbf{x}_i)(1 - p(y_i|\mathbf{x}_i)), & j = y_i \\ -p(y_i|\mathbf{x}_i)p(j|\mathbf{x}_i), & j \neq y_i \end{cases} \tag{11}$$

## B.2 DERIVATION OF CCE

According to Eq. (2), we have

$$L_{\text{CCE}}(\mathbf{x}_i; f_\theta, \mathbf{W}) = -\log p(y_i|\mathbf{x}_i). \tag{12}$$

Therefore, we obtain (the parameters are omitted for brevity),

$$\frac{\partial L_{\text{CCE}}}{\partial p(j|\mathbf{x}_i)} = \begin{cases} -p(y_i|\mathbf{x}_i)^{-1}, & j = y_i \\ 0, & j \neq y_i \end{cases}. \tag{13}$$

## B.3 DERIVATION OF MAE

According to Eq. (3), we have

$$L_{\text{MAE}}(\mathbf{x}_i; f_\theta, \mathbf{W}) = 2(1 - (p(y_i|\mathbf{x}_i))). \tag{14}$$

Therefore, we obtain

$$\frac{\partial L_{\text{MAE}}}{\partial p(j|\mathbf{x}_i)} = \begin{cases} -2, & j = y_i \\ 0, & j \neq y_i \end{cases}. \tag{15}$$

## B.4 DERIVATION OF GCE

According to Eq. (4), we have

$$L_{\text{GCE}}(\mathbf{x}_i; f_\theta, \mathbf{W}) = \frac{1 - p(y_i|\mathbf{x}_i)^q}{q}. \tag{16}$$

Therefore, we obtain

$$\frac{\partial L_{\text{GCE}}}{\partial p(j|\mathbf{x}_i)} = \begin{cases} -p(y_i|\mathbf{x}_i)^{q-1}, & j = y_i \\ 0, & j \neq y_i \end{cases}. \tag{17}$$

## B.5 DERIVATIVES W.R.T. LOGITS $\mathbf{z}_i$

### B.5.1 $\partial L_{\text{CCE}}/\partial \mathbf{z}_i$

The calculation is based on Eq. (13) and Eq. (11).

If $j = y_i$, we have:

$$\frac{\partial L_{\text{CCE}}}{\partial \mathbf{z}_{iy_i}} = \sum_{j=1}^{C} \frac{\partial L_{\text{CCE}}}{\partial p(j|\mathbf{x}_i)} \frac{\partial p(y_i|\mathbf{x}_i)}{\mathbf{z}_{ij}} \tag{18}$$
$$= p(y_i|\mathbf{x}_i) - 1.$$

If $j \neq y_i$, it becomes:

$$\frac{\partial L_{\text{CCE}}}{\partial \mathbf{z}_{ij}} = \sum_{j=1}^{C} \frac{\partial L_{\text{CCE}}}{\partial p(j|\mathbf{x}_i)} \frac{\partial p(y_i|\mathbf{x}_i)}{\mathbf{z}_{ij}} \tag{19}$$
$$= p(j|\mathbf{x}_i).$$

*In summary*, $\partial L_{\text{CCE}}/\partial \mathbf{z}_i$ can be represented as:

$$\frac{\partial L_{\text{CCE}}}{\partial \mathbf{z}_{ij}} = \begin{cases} p(y_i|\mathbf{x}_i) - 1, & j = y_i \\ p(j|\mathbf{x}_i), & j \neq y_i \end{cases}. \tag{20}$$

### B.5.2 $\partial L_{\mathrm{MAE}}/\partial \mathbf{z}_i$

The calculation is analogous with that of $\partial L_{\mathrm{CCE}}/\partial \mathbf{z}_i$.

According to Eq. (15) and Eq. (11), if $j = y_i$:

$$
\begin{aligned}
\frac{\partial L_{\mathrm{MAE}}}{\partial \mathbf{z}_{iy_i}} &= \sum_{j=1}^{C} \frac{\partial L_{\mathrm{MAE}}}{\partial p(j|\mathbf{x}_i)} \frac{\partial p(y_i|\mathbf{x}_i)}{\mathbf{z}_{ij}} \\
&= -2p(y_i|\mathbf{x}_i)(1 - p(y_i|\mathbf{x}_i)).
\end{aligned}
\tag{21}
$$

otherwise ($j \neq y_i$):

$$
\begin{aligned}
\frac{\partial L_{\mathrm{MAE}}}{\partial \mathbf{z}_{ij}} &= \sum_{j=1}^{C} \frac{\partial L_{\mathrm{MAE}}}{\partial p(j|\mathbf{x}_i)} \frac{\partial p(y_i|\mathbf{x}_i)}{\mathbf{z}_{ij}} \\
&= 2p(y_i|\mathbf{x}_i)p(j|\mathbf{x}_i).
\end{aligned}
\tag{22}
$$

*In summary*, $\partial L_{\mathrm{MAE}}/\partial \mathbf{z}_i$ is:

$$
\frac{\partial L_{\mathrm{MAE}}}{\partial \mathbf{z}_{ij}} = \begin{cases} 2p(y_i|\mathbf{x}_i)(p(y_i|\mathbf{x}_i) - 1), & j = y_i \\ 2p(y_i|\mathbf{x}_i)p(j|\mathbf{x}_i), & j \neq y_i \end{cases}.
\tag{23}
$$

### B.5.3 $\partial L_{\mathrm{GCE}}/\partial \mathbf{z}_i$

The calculation is based on Eq. (17) and Eq. (11).

If $j = y_i$, we have:

$$
\begin{aligned}
\frac{\partial L_{\mathrm{GCE}}}{\partial \mathbf{z}_{iy_i}} &= \sum_{j=1}^{C} \frac{\partial L_{\mathrm{GCE}}}{\partial p(j|\mathbf{x}_i)} \frac{\partial p(y_i|\mathbf{x}_i)}{\mathbf{z}_{ij}} \\
&= p(y_i|\mathbf{x}_i)^q(p(y_i|\mathbf{x}_i) - 1).
\end{aligned}
\tag{24}
$$

If $j \neq y_i$, it becomes:

$$
\begin{aligned}
\frac{\partial L_{\mathrm{GCE}}}{\partial \mathbf{z}_{ij}} &= \sum_{j=1}^{C} \frac{\partial L_{\mathrm{GCE}}}{\partial p(j|\mathbf{x}_i)} \frac{\partial p(y_i|\mathbf{x}_i)}{\mathbf{z}_{ij}} \\
&= p(y_i|\mathbf{x}_i)^q p(j|\mathbf{x}_i).
\end{aligned}
\tag{25}
$$

*In summary*, $\partial L_{\mathrm{GCE}}/\partial \mathbf{z}_i$ can be represented as:

$$
\frac{\partial L_{\mathrm{GCE}}}{\partial \mathbf{z}_{ij}} = \begin{cases} p(y_i|\mathbf{x}_i)^q(p(y_i|\mathbf{x}_i) - 1), & j = y_i \\ p(y_i|\mathbf{x}_i)^q p(j|\mathbf{x}_i), & j \neq y_i \end{cases}.
\tag{26}
$$

## C  Small-scale Fine-grained visual categorisation of vehicles

How does GR perform on small datasets, for example, the number of data points is no more than 5,000? We have tested GR on CIFAR-10 and CIFAR-100 in the main paper. However, both of them contain a training set of 50,000 images.

For this question, we answer it from different perspectives as follows:

*1. The problem of label noise we study on CIFAR-10 and CIFAR-100 in Section 4.2 is of similar scale.* For example:

- In Table 4, when noise rate is 80% on CIFAR-10, the number of clean training examples is around $50,000 \times 20\% = 5,000 \times 2$. Therefore, this clean set is only two times as large as 5,000. Beyond, the learning process may be interrupted by other noisy data points.

- In Table 5, when noise rate is 60% on CIFAR-100, the number of clean training data points is about $50,000 \times 40\% = 5,000 \times 4$, i.e., four times as large as 5,000.

*2. We compare GR with other standard regularisers on a small-scale fine-grained visual categorisation problem in Table 9.*

**Vehicles-10 Dataset.** In CIFAR-100 Krizhevsky (2009), there are 20 coarse classes, including vehicles 1 and 2. Vehicles 1 contains 5 fine classes: bicycle, bus, motorcycle, pickup truck, and train. Vehicles 2 includes another 5 fine classes: lawn-mower, rocket, streetcar, tank, and tractor. We build a small-scale vehicles classification dataset composed of these 10 vehicles from CIFAR-100. Specifically, the training set contains 500 images per vehicle class while the testing set has 100 images per class. Therefore, the number of training data points is 5,000 in total.

Table 9: The test accuracy (%) of GR and other standard regularisers on Vehicles-10. We train ResNet-44. Baseline means CCE without regularisation. We test two cases: with symmetric label noise $r = 40\%$ and without symmetric label noise $r = 0$.

| $r$ | Baseline | L2 | Dropout | Dropout+L2 | GR | GR+L2 | GR+Dropout | GR+L2+Dropout |
|---|---|---|---|---|---|---|---|---|
| 0 | 75.4 | 76.4 | 77.9 | 78.7 | 83.8 | 84.4 | 84.5 | **84.7** |
| 40% | 42.3 | 44.8 | 41.6 | 47.4 | 45.8 | 55.7 | 48.8 | **58.1** |

## D  TRAINING UNDER ASYMMETRIC LABEL NOISE

We evaluate on CIFAR-100, whose 100 classes are grouped into 20 coarse classes. Every coarse class has 5 fine classes. Within each coarse class, an image's label is flipped to one of the other four labels uniformly with a probability $r$. $r$ represents the noise rate. We set $r = 0.2$. The results are displayed in Table 10. When GR is used, the performance is better than its counterparts without GR.

Table 10: The test accuracy (%) of GR and other standard regularisers trained under asymmetric label noise. We train ResNet-44. Baseline means CCE without regularisation. We simply fix $\beta = 8, \lambda = 0.5$ when GR is used. Better results can be expected if $\beta, \lambda$ are optimised for each case.

| Baseline | L2 | Dropout | Dropout+L2 | GR | GR+L2 | GR+Dropout | GR+L2+Dropout |
|---|---|---|---|---|---|---|---|
| 55.1 | 59.4 | 57.1 | 60.4 | 60.5 | **63.7** | 59.4 | 61.4 |

## E  THE EFFECTIVENESS OF LABEL CORRECTION

The results are shown in Table 11.

Table 11: How much fitting of the clean training subset and how much fitting of the noisy training subset? Is it plausible to correct the labels of training data?
Our results demonstrate the effectiveness of label correction using DNNs trained by GR.
When retraining from scratch on the relabelled training data, we do not adjust the hyper-parameters $\beta$ and $\lambda$. Therefore, the reported results of retraining on relabelled datasets are not the optimal.

| Noise Rate $r$ | Emphasis Focus | Model | Testing Accuracy (%) | | Accuracy on Training Sets (%) | | Fitting degree of subsets (%) | | Retrain after label correction |
|---|---|---|---|---|---|---|---|---|---|
| | | | Best | Final | Noisy | Intact | Clean | Noisy | |
| 20% | 0 | CCE | 86.5 | 76.8 | **95.7** | 80.6 | 99.0 | 85.9 | – |
| | 0~0.5 ($\lambda = 0.5$) | GR ($\beta = 12$) | **89.4** | **87.8** | 81.5 | **95.0** | 98.8 | 11.7 | 89.3 (+1.5) |
| 40% | 0 | CCE | 82.8 | 60.9 | **83.0** | 64.4 | 97.0 | 81.1 | – |
| | 0.5 ($\lambda = 1$) | GR ($\beta = 16$) | 84.7 | **83.3** | 60.3 | **88.9** | 94.8 | 7.5 | 85.3 (+2) |

# F MORE EMPIRICAL RESULTS

## F.1 REVIEW

**Question**: What training examples should be focused on and how much more should they be emphasised when training DNNs under label noise?

**Proposal**: Gradient rescaling incorporates emphasis focus (centre/focal point) and emphasis spread, and serves as explicit regularisation in terms of sample reweighting/emphasis.

**Finding**: When noise rate is higher, we can improve a model's robustness by moving emphasis focus towards relatively less difficult examples.

## F.2 DETAILED RESULTS ON CIFAR-100

The more detailed results on CIFAR-100 are shown in Table 12, which is the supplementary of Table 5 in the main text.

Table 12: Exploration of GR with *different emphasis focuses (centres) and spreads* on CIFAR-100 when $r = 20\%, 40\%, 60\%$, respectively. *This table presents detailed information of optimising $\lambda, \beta$ mentioned in Table 5 in the paper.* Specifically, for each $\lambda$, we try 5 $\beta$ values from $\{2, 4, 6, 8, 10\}$ and select the best one as the final result of the $\lambda$. We report the mean test accuracy over 5 repetitions. Our key finding is demonstrated again: When $r$ raises, we can increase $\beta, \lambda$ for better robustness. The increasing scale is much smaller than CIFAR-10. This is because CIFAR-100 has 100 classes so that its distribution of $p_i$ (input-to-label relevance score) is different from CIFAR-10 after softmax normalisation.

| Noise rate $r$ | $\lambda$ | $\beta$ | Testing accuracy (%) |
|---|---|---|---|
| | 0.1 | 4 | 61.3 |
| | 0.2 | 4 | 63.3 |
| 20% | 0.3 | 6 | **64.1** |
| | 0.4 | 6 | 63.6 |
| | 0.5 | 8 | 62.6 |
| | 0.6 | 8 | 62.5 |
| | 0.1 | 4 | 55.5 |
| | 0.2 | 4 | 58.2 |
| 40% | 0.3 | 6 | 59.1 |
| | 0.4 | 6 | **60.0** |
| | 0.5 | 8 | 59.3 |
| | 0.6 | 8 | 58.5 |
| | 0.1 | 4 | 44.9 |
| | 0.2 | 4 | 47.5 |
| 60% | 0.3 | 6 | 49.7 |
| | 0.4 | 6 | **49.9** |
| | 0.5 | 8 | **49.9** |
| | 0.6 | 8 | 47.3 |

## F.3 DETAILED TRAINING DYNAMICS

There are more detailed training dynamics displayed in the Figures 4-10.

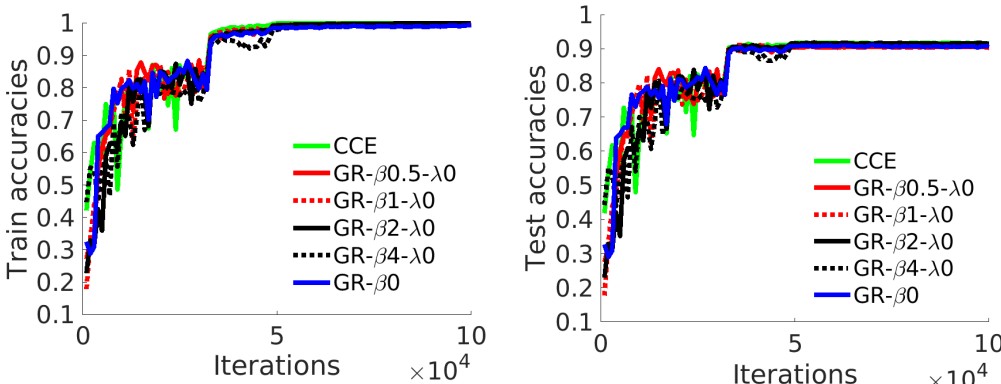

Figure 4: The training and test accuracies on clean CIFAR-10 along with training iterations. The training labels are clean. We fix $\lambda = 0$ to focus on harder examples while changing emphasis spread controller $\beta$. *The backbone is ResNet-20.* The results of ResNet-56 are shown in Figure 5. *Better viewed in colour.*

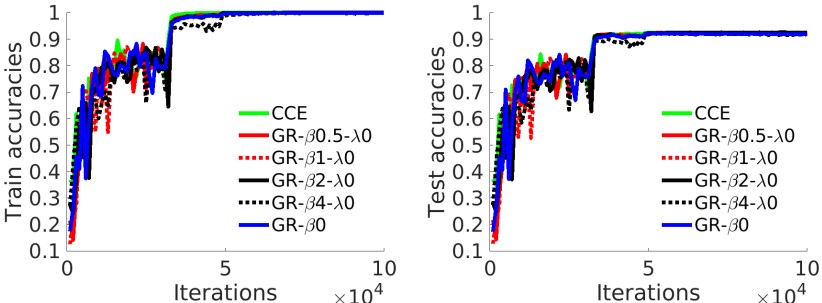

Figure 5: The training and test accuracies on clean CIFAR-10 along with training iterations. The training labels are clean. We fix $\lambda = 0$ to focus on more difficult examples while changing emphasis spread controller $\beta$. *The backbone is ResNet-56.* The results of ResNet-20 are shown in Figure 4. *Better viewed in colour.*

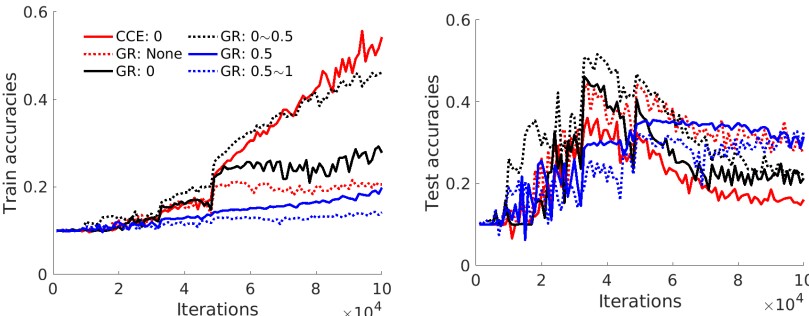

Figure 6: The learning dynamics on CIFAR-10 ($r = 80\%$) with ResNet-56, i.e., training and testing accuracies along with training iterations. The legend in the top left is shared by two subfigures. 'xxx: yyy' means 'method: emphasis focus'. The results of $r = 20\%, 40\%, 60\%$ are shown in Figure 2 in the paper.

We have two key observations: 1) When noise rate increases, better generalisation is obtained with higher emphasis focus, i.e., focusing on relatively easier examples; 2) Both overfitting and underfitting lead to bad generalisation. For example, 'CCE: 0' fits training data much better than the others while 'GR: None' generally fits it unstably or a lot worse. *Better viewed in colour.*

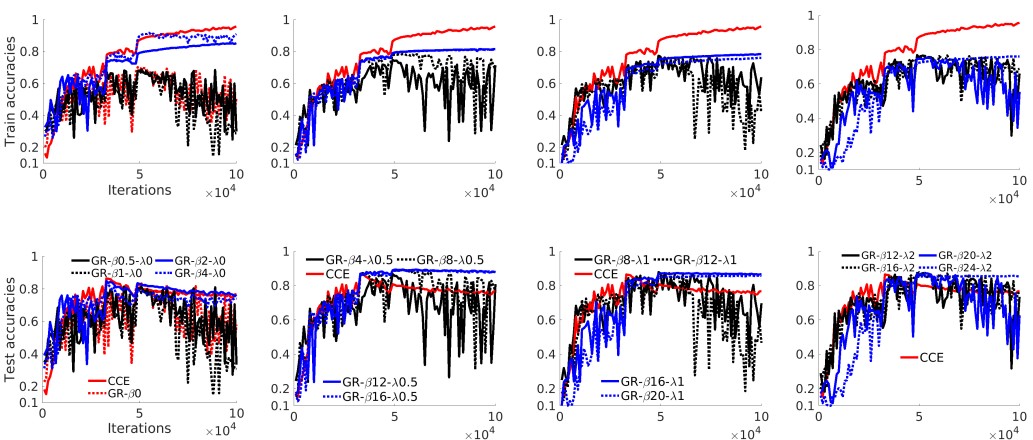

Figure 7: ResNet-56 on CIFAR-10 ($r = 20\%$). From left to right, the results of four emphasis focuses 0, 0~0.5, 0.5, 0.5~1 with different emphasis spreads are displayed in each column respectively. When $\lambda$ is larger, $\beta$ should be larger as displayed in Figure 1c in the paper. Specifically :
1) when $\lambda = 0$: we tried $\beta = 0.5, 1, 2, 4$;
2) when $\lambda = 0.5$: we tried $\beta = 4, 8, 12, 16$;
3) when $\lambda = 1$: we tried $\beta = 8, 12, 16, 20$;
4) when $\lambda = 2$: we tried $\beta = 12, 16, 20, 24$.

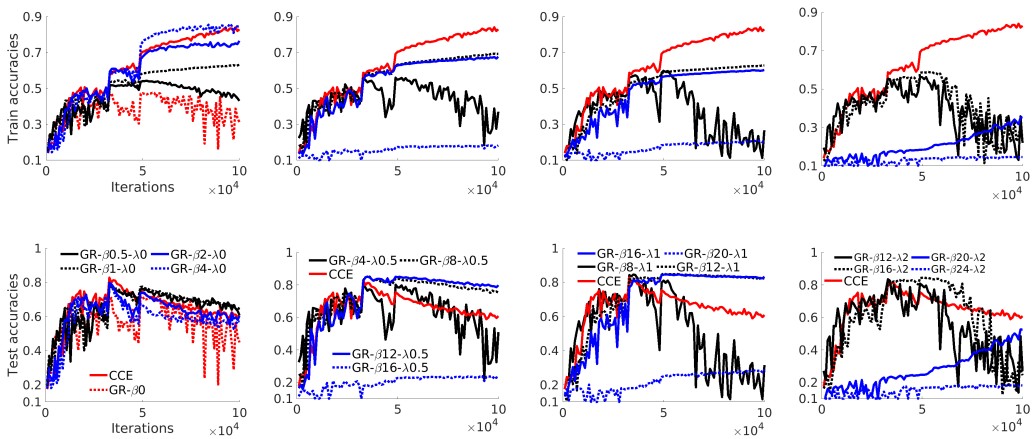

Figure 8: ResNet-56 on CIFAR-10 ($r = 40\%$). From left to right, the results of four emphasis focuses 0, 0~0.5, 0.5, 0.5~1 with different emphasis spreads are displayed in each column respectively. When $\lambda$ is larger, $\beta$ should be larger as displayed in Figure 1c in the paper. Specifically :
1) when $\lambda = 0$: we tried $\beta = 0.5, 1, 2, 4$;
2) when $\lambda = 0.5$: we tried $\beta = 4, 8, 12, 16$;
3) when $\lambda = 1$: we tried $\beta = 8, 12, 16, 20$;
4) when $\lambda = 2$: we tried $\beta = 12, 16, 20, 24$.

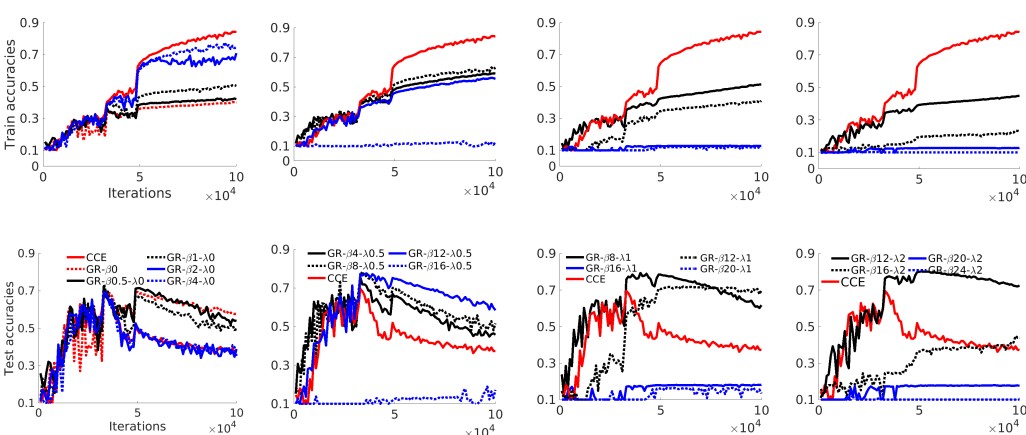

Figure 9: ResNet-56 on CIFAR-10 ($r = 60\%$). From left to right, the results of four emphasis focuses 0, 0~0.5, 0.5, 0.5~1 with different emphasis spreads are displayed in each column respectively. When $\lambda$ is larger, $\beta$ should be larger as displayed in Figure 1c in the paper. Specifically :
1) when $\lambda = 0$: we tried $\beta = 0.5, 1, 2, 4$;
2) when $\lambda = 0.5$: we tried $\beta = 4, 8, 12, 16$;
3) when $\lambda = 1$: we tried $\beta = 8, 12, 16, 20$;
4) when $\lambda = 2$: we tried $\beta = 12, 16, 20, 24$.

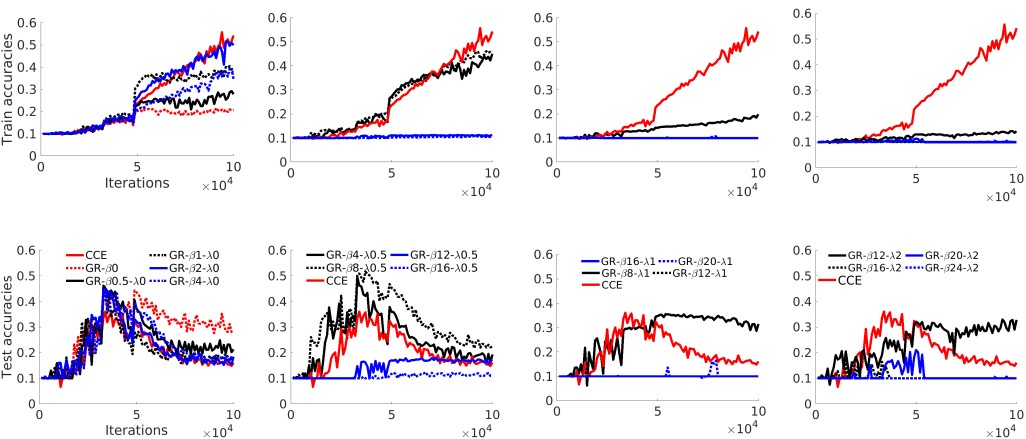

Figure 10: ResNet-56 on CIFAR-10 ($r = 80\%$). From left to right, the results of four emphasis focuses 0, 0∼0.5, 0.5, 0.5∼1 with different emphasis spreads are displayed in each column respectively. When $\lambda$ is larger, $\beta$ should be larger as displayed in Figure 1c in the paper. Specifically :
1) when $\lambda = 0$: we tried $\beta = 0.5, 1, 2, 4$;
2) when $\lambda = 0.5$: we tried $\beta = 4, 8, 12, 16$;
3) when $\lambda = 1$: we tried $\beta = 8, 12, 16, 20$;
4) when $\lambda = 2$: we tried $\beta = 12, 16, 20, 24$.

