# OpenReview forum: "ROBUST DISCRIMINATIVE REPRESENTATION LEARNING VIA GRADIENT RESCALING: AN EMPHASIS REGULARISATION PERSPECTIVE"
_ICLR.cc/2020/Conference — Reject_

### Official Review · AnonReviewer1 · 2019-10-23
**Official Blind Review #1**

**Rating:** 6

**Review:**

Summary:
The paper proposes a method for noise robustness based on scaling gradients of examples. By choosing the proper scaling parameters (alpha and beta), the method recovers standard losses such as CCE, MAE, and GCE, while also recovering other losses. The method is strongly related to reweighting training examples, where alpha and beta define the shape of this weighting as a function of the model's prediction (i.e., p_i). Experiments show that the proposed method achieves competitive results on several standard benchmarks for noisy-labelled data.

Comments:
- The main drawback of the proposed approach is that there is no clear way of choosing alpha and beta, other than a hyper-parameter search, which is not very practical and can lead to overfitting the test set.
- The paper in general is easy to follow, but the paper is not very rigorous or clear on some important concepts. For example:
     * No clear mathematical definition of emphasis focus and spread
     * The term "semantically abnormal examples" should be defined in the main text.
     * It is not so clear what it means to "babyset" emphasis focus and spread.
     * I don't understand what Eq. 6 is supposed to tell.
     * What are the \dots in equation
- The experiments are very thorough and the results are very good, but I have few clarifying questions:
     * The procedure for choosing beta/gamma is not clear, and I see that for every experiment those values change.
     * It would be nice if the CIFAR-10 Table 4 results are performed using the exact same setting to prior work to make sure the comparison is fair. For example, GCE results in (Zhang & Sabuncu 2018) are much that the reported ones. While it seems that you're using GoogLeNet V1 architecture similar to Jiang et al. 2018, it's not clear which experimental setting you are comparing against.
     * Can you be more specific what do you mean by "with a little effort for optimizing beta and gamma" in caption of Table 5?

Minor:
     * Grammer mistake: "what training examples...focused *on*..."
     * Citations should be done with parentheses


**Experience Assessment:**

I have read many papers in this area.

**Review Assessment: Checking Correctness Of Derivations And Theory:**

I assessed the sensibility of the derivations and theory.

**Review Assessment: Checking Correctness Of Experiments:**

I assessed the sensibility of the experiments.

**Review Assessment: Thoroughness In Paper Reading:**

I read the paper thoroughly.

---

> ### Author Response · Authors · 2019-11-07
> **Procedure and Principle for choosing beta/lambda, Experimental setup and Others**
>
>
> 1. The main drawback of the proposed approach is that there is no clear way of choosing alpha and beta, other than a hyper-parameter search, which is not very practical and can lead to overfitting the test set.
> 2. The experiments are very thorough and the results are very good, but I have few clarifying questions:
>     2.1. The procedure for choosing beta/gamma is not clear, and I see that for every experiment those values change.
>     2.2. Can you be more specific what do you mean by "with a little effort for optimizing beta and gamma" in caption of Table 5?
>
> We reply to questions 1 and 2 together as they are related.
>
> - Reply to 1 & 2.1 on “the procedure for choosing beta/lambda”:
> We have summarised the procedure and principle for choosing $\lambda,\beta$ in the empirical analysis part, i.e., Section 4.2.1. Details are as follows:
>   (1)  Emphasis focus: When noise rate is higher, we can improve a model’s robustness by moving emphasis focus towards relatively less difficult examples with a larger $\lambda$, which is informative in practice.
>   (2)  When $\lambda$ is larger, $\beta$ should be larger as shown in Figure 1c. This is also demonstrated in Table 3, Table 5, and Table 10.
>
> - Reply to 2.1 on “for every experiment those values change”:
> In Table 4 and Table 5, we have displayed the results of GR with fixed parameters $(\beta=8, \lambda=0.5)$. Those results are better than the state-of-the-art.
>
> - Reply to 2.2:
> Those empirical results above make it easy to optimise $\lambda,\beta$ in practice. Therefore, we wrote “with a little effort for optimizing $\lambda,\beta$”.
>
>
> 3. It would be nice if the CIFAR-10 Table 4 results are performed using the exact same setting to prior work to make sure the comparison is fair. For example, GCE results in (Zhang & Sabuncu 2018) are much that the reported ones. While it seems that you're using GoogLeNet V1 architecture similar to Jiang et al. 2018, it's not clear which experimental setting you are comparing against.
>
> - The research problem we work on is important and popular. Therefore, on different datasets, different baselines along with different net architectures have been evaluated in the literature. For example, some prior work reimplemented baselines with their custom-designed net architectures. The experimental settings, e.g., choice of net architectures, are not consistent and rigid in different prior work. On the one hand, it is nice since it provides diverse evaluation settings. On the other hand, it presents a challenge to keep the compared baselines consistent when you aim to compare fairly with different types of baselines on different datasets. Especially some prior work designed their own net architectures.
>
> - In our experimental setup, we aim to test on different benchmarks using different publicly available net architectures for more comprehensive evaluation:
>   (1) We choose GoogLeNet V1 on CIFAR-10 following MentorNet (Jiang et al., 2018), where most baselines are example reweighting algorithms as shown in Table 4.
>   (2) We use ResNet-44 on CIFAR-100 following D2L (Ma et al., 2018), where most competitors are specifically designed for addressing label noise as displayed in Table 5.
>   (3) The experiments on Clothing 1M using ResNet-50 are consistent in most prior work. We have displayed most related baselines in Table 6.
>
> - We remark that on each dataset, we only report the results of those baselines using the same net architecture for a fair comparison.
>
>
>
> 4. The paper in general is easy to follow, but the paper is not very rigorous or clear on some important concepts. For example:
>
>   4.1 It is not so clear what it means to "babysit" emphasis focus and spread.
>   It means that we need to choose emphasis focus and spread properly instead of using the built-in default settings in loss functions.
>
>   4.2 I don't understand what Eq. 6 is supposed to tell.
>   We use Eq. (6) to tell that GR can be independent of empirical loss formulations. Since gradient computation is independent of loss computation, different losses only indicate how far we are away from different minimisation objectives. The supervision information of GR can be controlled independently and straightforwardly.
>
>   4.3 & 4.4 No clear mathematical definition of emphasis focus and spread. What are the \dots in equation?
>   We have provided mathematical definitions of emphasis focus and spread in our revised version.
>   We have removed those \cdot in our revised version.
>
>
> 5. Minor:
>      * Grammer mistake: "what training examples...focused *on*..."
>      * Citations should be done with parentheses
>
>   Thanks so much for your helpful and careful check. We have revised them in our revised version.

---

> > ### Comment · AnonReviewer1 · 2019-11-14
> > **Thanks for your response**
> >
> > Thanks for addressing my questions.

---

### Official Review · AnonReviewer2 · 2019-10-26
**Official Blind Review #2**

**Rating:** 3

**Review:**

Summary
This paper presents Gradient Rescaling (GR) for robust learning to combat label noise. They propose to treat each data sample with different significance scores: some samples are important to learning, and some examples are insignificant (or even detrimental) to learning. So they desire to weight each samples according to their significance. They propose the notion of emphasis focus (When learning, whether we should put emphasis on learning “hard” examples or “easy” examples) and emphasis spread (the variance of these significance weights). The authors propose that this “difficulty” of samples are proportional to their network output logit values.
The authors examine the analytical forms of the gradients of popular loss functions such as Categorical Cross Entropy, Mean Absolute Error and Generalized Cross Entropy. They find that the formulas for the gradient are of similar family with varying hyperparameters. Authors claim that tweaking these hyperparameters result in tuning the emphasis focus and spread.
The authors conduct Experiments on CIFAR10, CIFAR100 with simulated symmetric noise. Also, they conduct experiments on real-world noisy datasets: Clothing 1M dataset and MARS video dataset. The authors claim that the performance of GR exceeds various baselines.


Significance/Novelty/Clarity
Significance: Low-Medium. The performance increase exhibited in the experiments are a bit underwhelming (when considering the fact that benchmarks of most recent noise-robustness algos such as <Lee et al. 2019 ICML> are missing).
Novelty: Medium. The paper is interesting in the sense that the authors integrated (and allegedly generalized) the gradient formulas for several losses into one family, and tried to integrate and tweak their postulation of  “Emphasis focus” and “emphasis spread” into the framework. However, the theoretical ground and convincing reasoning for their claim seems a bit lackluster.
Clarity: Low. The overall flow of the paper is a bit fuzzy - exhibiting a stream-of-consciousness style flow.


Pros and Cons in Detail
Pros:
1.The authors try to unify the analytical forms of the gradients of various loss functions into a single family equipped with hyperparameters that control emphasis focus and spread.
2.Conducted experiments show that GR achieved increased performance when compared to the baselines.
Cons:
My major concern is about tuning newly introduced hyperparameters in practical settings. How can we guarantee to have intact validation set?  Can we get any improvement via GR even with corrupted validation set for tuning hyperparameters?
1. The arguments of the authors are grounded in the premise that “difficult” samples will exhibit small logit values, and “easy” samples high logit values.
2. No justifications (both theoretical and experimental) are provided on the claim that controlling emphasis focus/spread will result in more robust learning.
3. This algorithm introduces 2 additional hyperparameters that are correlated with each other. This introduces additional labor.
4. By changing the loss function, the outputs of the network might lose its interpretation as a probability distribution.
5. No confidence intervals are shown except for the CIFAR-100 experiment.
6. Experiments are only conducted on vision tasks.
7. The baseline menagerie also changes when the authors change the target dataset.
8. Additional benchmarks of most recent noise-robustness algos such as <Lee et al. 2019 ICML> are required.


Questions
1. Is it always the case that “difficult” samples exhibit small logit values, and “easy” samples high logit values?
2. If not, GR’s emphasis manipulation might result in neglecting samples containing valuable information.
3. Can GR be used simultaneously with other noise-robust learning methods to further boost the performance?
4. Technically, GR aims to rescale the gradients of the logits. How will it interact with optimizers  other than SGD such as Adam?
5. Does GR still work well on small datasets(#points < 5000)?


Misc. Comments
Page 3-> inside L1 norm, no differentiation sign in the denominator.
Around eq 2 and 4: missing derivative symbol w.r.t. z

**Experience Assessment:**

I have read many papers in this area.

**Review Assessment: Checking Correctness Of Derivations And Theory:**

I assessed the sensibility of the derivations and theory.

**Review Assessment: Checking Correctness Of Experiments:**

I assessed the sensibility of the experiments.

**Review Assessment: Thoroughness In Paper Reading:**

I read the paper at least twice and used my best judgement in assessing the paper.

---

> ### Author Response · Authors · 2019-11-15
> **Questions**
>
>
> Thank you for your questions.
>
> 1&2: Is it always the case that “difficult” samples exhibit small logit values, and “easy” samples high logit values? If not,.....
> Generally, the answer is yes. As training goes, the premise that semantic anomalies have small classification confidences while normal examples tend to have large classification confidences is indeed our reasonable assumption.
> Actually, a lot of prior work has demonstrated the reasonability of this premise. In addition, many algorithms have been proposed based on similar premises. Our empirical analysis also supports this premise well. More details are as follows:
>
> (1) Technically, we have also remarked that “we do not design the weighting scheme heuristically from scratch. Instead, it is naturally motivated by the gradient analysis of several loss functions”, which makes GR principally and technically sound. More discussion is provided in Section 3.3.
>
> (2) In terms of prior work, self-paced learning, e.g., Self-paced (Kumar et al., 2010), and curriculum learning, e.g., MentorNet (Jiang et al., 2018) are practical algorithms based on this premise. In addition, as demonstrated in (Krueger et al., 2017; Arpit et al., 2017), when severe noise exists, DNNs learn simple meaningful patterns first before memorising abnormal examples.
>
> (3) Finally, in the “emphasis focus” paragraph of Section 1, we discussed that “It is a common practice to focus on harder instances when training DNNs (Shrivastava et al., 2016; Lin et al., 2017).”
> “When a dataset is clean, it achieves faster convergence and better performance to emphasise on harder examples because they own larger gradient magnitude, which means more information and a larger update step for model’s parameters.”
> “However, when severe noise exists, as demonstrated in (Krueger et al., 2017; Arpit et al., 2017), DNNs learn simple meaningful patterns first before memorising abnormal ones.”
> “In other words, anomalies are harder to fit and own larger gradient magnitude in the later stage. Consequently, if we use the default sample weighting in categorical cross entropy (CCE) where harder samples obtain higher weights, anomalies tend to be fitted well especially when a network has large enough capacity. That is why we need to move the emphasis focus towards relatively easier ones, which serves as emphasis regularisation.”
>
> 3. Can GR be used simultaneously with other noise-robust learning methods to further boost the performance?
> We have tried combining GR with other standard regularisers in Section 4.5, please see Table 8. For example, $\mathit{Dropout \ is \ demonstrated \ to \ be \ a \ great \ regulariser \ against \ label \ noise \ in \ Arpit \ et \ al., \ 2017}$, “A closer look at memorization in deep networks”. This is also demonstrated in our Table 8, e.g., Dropout > Baseline, and Dropout+L2 > L2.
>
> Case 1: GR can help other regularisation techniques.
> GR consistently improves the generalisation performance after it is added:
> GR > Baseline;
> GR+L2 > L2;
> GR+Dropout > Dropout;
> GR+L2+Dropout > Dropout+L2.
> Case 2: Other regularisation techniques may not help GR.
> When GR is already applied, adding another regulariser may not lead to better regularisation effect and generalisation performance. For example:
> Adding Dropout hurts the performance:
> GR+Dropout < GR;
> GR+L2+Dropout < GR+L2.
> Adding L2 decay improves the performance:
> GR+L2 > GR;
> GR+L2+Dropout > GR+Dropout.
>
> Therefore, we are sorry that there is no deterministic answer for your question. The search and studying space is large when considering the diverse combination options of different regularisers.
> However, it is worth noting that the interaction over multiple regularisers may not improve the generalisation performance in practice.
>
> 5. Does GR still work well on small datasets(#points < 5000)?
> To address your concern on this question, we add a Section  $C$ in our supplementary material in the new revised version. Please have a check. Thanks.
> (1). The problem of label noise we study on CIFAR-10 and CIFAR-100 in Section 4.2 is of similar scale.
> (2). We compare GR with other standard regularisers on a small-scale fine-grained visual categorisation problem in Table 9. The number of training data points is 5,000 in total.
> This new experiment demonstrates that GR works on small datasets as well.

---

> ### Author Response · Authors · 2019-11-15
> **Cons 5-8**
>
>
> 5. No confidence intervals are shown except for the CIFAR-100 experiment.
> Sorry, we do not understand this question, could you please explain more in detail?
>
>
> 6. Experiments are only conducted on vision tasks.
> We analyse and demonstrate GR’s effectiveness on diverse computer vision tasks using different net architectures:
> 1) Image classification with clean training data, e.g., CIFAR-10 and CIFAR-100 datasets;
> 2) Image classification with synthetic symmetric label noise, which is more challenging than asymmetric noise evaluated by (Vahdat, 2017; Ma et al., 2018), e.g., CIFAR-10 and CIFAR-100 datasets;
> 3) Image classification with real-world unknown anomalies, which may contain open-set noise (Wang et al., 2018), e.g., images with only background, or outliers, etc. We test on Clothing 1M dataset;
> 4) Video person re-identification, a video retrieval task containing diverse anomalies, e.g., MARS dataset.
> 5) We show that GR is notably better than other standard regularisers, e.g., L2 weight decay and dropout. Besides, to comprehensively understand GR’s behaviours, we present extensive ablation studies.
>
> We agree it is always more convincing to evaluate on more tasks. We are working on it.
>
>
> 7. The baseline menagerie also changes when the authors change the target dataset.
> * Background on experimental settings: The research problem we work on is important and popular. Therefore, on different datasets, different baselines along with different net architectures have been evaluated in the literature. For example, some prior work reimplemented baselines with their custom-designed net architectures. The experimental settings, e.g., choice of net architectures, are not consistent and rigid in different prior work. On the one hand, it is nice since it provides diverse evaluation settings. On the other hand, it presents a challenge to keep the compared baselines consistent when you aim to compare fairly with different types of baselines on different datasets. Especially some prior work designed their own net architectures.
>
> * In our experimental setup, we aim to test on different benchmarks using different publicly available net architectures for more comprehensive evaluation:
> 1). We choose GoogLeNet V1 on CIFAR-10 following MentorNet (Jiang et al., 2018), where most baselines are example reweighting algorithms as shown in Table 4.
> 2). We use ResNet-44 on CIFAR-100 following D2L (Ma et al., 2018), where most competitors are specifically designed for addressing label noise as displayed in Table 5.
> 3). The experiments on Clothing 1M using ResNet-50 are consistent in most prior work. We have displayed most related baselines in Table 6.
>
> * We remark that on each dataset, we only report the results of those baselines using the same net architecture for a fair comparison.
>
>
> 8. Additional benchmarks of most recent noise-robustness algos such as <Lee et al. 2019 ICML> are required.
> We introduce the reasons for without benchmarking the Robust Inference method (Lee et al. 2019 ICML). They have a misstatement in their experimental section 4.3, making it hard to compare fairly. More details are as follows:
>
> In the experimental section 4.3 of Lee et al., 2019, it is stated they followed the same experimental setup as D2L (Ma et al., 2018). However, we were surprised by its reported much better performance than the original results reported in D2L (Ma et al., 2018).
> Therefore, we emailed the authors of this paper and got the reply that they used different networks in their implementation. They did not provide us with what exact net architecture they used.
>
> Furthermore, Lee et al., 2019 is orthogonal to our work. We target at robust learning during training. On the contrary, Lee et al., 2019 proposed an inference method, Robust Generative classifier (RoG), which is applicable to a discriminative neural classifier pre-trained on noisy datasets (without retraining).
> Their premise is that the softmax DNNs can learn meaningful feature patterns shared by multiple training examples even under datasets with noisy labels. However, this premise is not always true without proper regularisation as shown in our Table 3. A deep model trained by softmax and cross entropy is able to fit noisy datasets well and learn non-meaningful patterns. This is also demonstrated in prior work Rethinking generalisation (Zhang et al., 2017 ICLR).

---

> ### Author Response · Authors · 2019-11-15
> **Cons 2-4**
>
>
> 2. No justifications (both theoretical and experimental) are provided on the claim that controlling emphasis focus/spread will result in more robust learning.
> Empirically, we have presented extensive experimental analysis in Section 4.2.1 and supplementary material.
>
> Technically, we discussed some justifications in Section 2.1 and Section 3.3. We emphasize that:
> 1). In this work, inspired by the analysis of CCE, MAE and GCE, which only differ in the gradient magnitude while perform quite differently, leading to a natural interpretation that gradient magnitude matters. That is why we explore rescaling the gradient magnitude as illustrated in Figure 1 and Table 1.
> 2). GR is independent of empirical loss expressions as presented in Table 1. Therefore, one specific loss is merely an indicator of how far we are away from a specific minimisation objective. It has no direct impact on the robustness of the learning process since it has no direct influence on the gradient back-propagation.
> Similar to the prior work of rethinking generalisation (Zhang et al., 2017), we need to rethink robust training under diverse anomalies, where the robustness theorems conditioned on symmetric losses and label noise are not directly applicable.
>
> In summary, our work is more about new findings and analysis on robust learning against diverse semantic anomalies. Although it does not build new theorems, GR challenges existing robustness theorems conditioned on symmetric losses and label noise. New theorems need to be built in the future when it comes to diverse semantic anomalies.
>
>
> 3. This algorithm introduces 2 additional hyperparameters that are correlated with each other. This introduces additional labor.
> Frankly, this is a difficult issue to address as hyperparameters generally exist when a new regulariser/method is proposed. We provide our thinking as follows:
> (1) It is natural that emphasis focus and spread are correlated since the classification confidence is distributed in a bounded range [0, 1].
>
> (2) Regarding the additional labour, we introduce our understanding: In our framework, there are two regularisation concepts, emphasis focus and spread. In deep learning, whenever a new regulariser is proposed, we have to consider its regularisation weight in practice. And when two or more than two regularisers are combined together, they are correlated naturally.
>
> (3) We are also thinking on how to reduce the labour of optimising a model’s hyperparameters when training a deep model. One possible solution may be using AutoML techniques to optimize a model’s hyperparameters automatically.
>
>
> 4. By changing the loss function, the outputs of the network might lose its interpretation as a probability distribution.
> Sorry, we find it is a factual misunderstanding. In the loss layer, we do not apply automatic gradient computation and back-propagation because GR makes the forward loss computation and backward gradient computation independent.
> We do not change the loss computation as we merely regard it as an indicator of how far we are away from a specific minimisation objective. It has no direct impact on the robustness of the learning process since it has no direct influence on the gradient back-propagation. Please see Section 2.1 for more details.
>
> Therefore, you can choose any loss computation as you need. You can also compute and output multiple losses. All loss indicators displayed in Table 1, i.e., CCE, MAE and GR, are computed by taking a predicted probability distribution as input. The predicted probability distribution can be computed as normal even if the loss computation is changed.

---

> ### Author Response · Authors · 2019-11-15
> **The Major Concern and Misc. Comments**
>
>
> 1. The major concern: How can we guarantee to have intact validation set? Can we get any improvement via GR even with corrupted validation set for tuning hyperparameters?
> For those questions, we would like to clarify two core concepts: $Robust \ Learning \ when \ abnormal \ Training \ examples \ exist \ and \ Model \ Selection \ according \ to \ a  \ Clean \ Validation  \ set$.
>
> $Robust \ Learning$: In our context, we mean the robustness against diverse semantic anomalies in the training set. In other words, when there are anomalies in the training set, we aim to learn meaningful patterns without fitting those semantic anomalies. Fitting anomalies well will definitely hurt the robustness and generalisation performance because a model learns errors and non-meaningful patterns by fitting anomalies.
>
> $Model \ Selection$: Using a validation set to evaluate and decide which model is better is a common practice in machine learning. This is because we are only able to see the test data after deployment.
> When we compare different models, a constructed validation set has to be clean so that it can serve as an oracle. We cannot evaluate/decide a model’s performance according to a noisy validation set:
> (1) If an example has an incorrect label and its true label is unknown, it would be an error to evaluate whether its predicted label is the same as the incorrect label.
> (2) Another intuitive and straightforward example: Given a validation dataset with an unknown noise rate, $A$ model with 80% accuracy may be worse than $B$ model with 70% accuracy because $A$ model may predict those anomalies better, which means $A$ model makes more wrong decisions.
> (3) If an out-of-distribution example exists, since it does not belong to any training class in fact, what we should do is to conduct out-of-distribution example detection and reject predicting it.
>
>
> 2. Misc. Comments
> Page 3-> inside L1 norm, no differentiation sign in the denominator.
> Around eq 2 and 4: missing derivative symbol w.r.t. z
>
>
> Thank you so much for your careful and helpful check. We have revised them in our revised version.

---

### Official Review · AnonReviewer3 · 2019-10-27
**Official Blind Review #3**

**Rating:** 3

**Review:**

Summary:
The authors first analyze and answer the question: What training examples should be focused and how large the emphasis spread should be? Then, they proposed the gradient rescaling framework serving as emphasis regularization.

Strengths:
1. The paper is well organized except the reference citation (read difficultly)
2. The proposed method is very simple and effective.
3. Experiments show the improvements over SOTA.

Weakness:
1. The experiments lack the recent important baseline "symmetric cross entropy for robust learning with noisy labels, ICCV2019", which are the current SOTA. Maybe the author should check the above paper and show the results.
2. The experiments are only conducted on symmetric noise. Actually, asymmetric noise is also important. The author should conduct at least some experiments on asymmetric noise.

**Experience Assessment:**

I have published in this field for several years.

**Review Assessment: Checking Correctness Of Derivations And Theory:**

I carefully checked the derivations and theory.

**Review Assessment: Checking Correctness Of Experiments:**

I carefully checked the experiments.

**Review Assessment: Thoroughness In Paper Reading:**

I read the paper thoroughly.

---

> ### Author Response · Authors · 2019-11-06
> **Comparing with ICCV19 baseline-SL and Why not evaluate on asymmetric noise?**
>
>
> Thanks so much for your helpful review. We are glad that you like our simple, effective and principled method.
> Regarding the weakness points you mentioned, we clarify them as follows. We look forward to further discussion with you.
>
> 1. Regarding the recent baseline “symmetric cross entropy for robust learning with noisy labels, ICCV2019”, we read it before submission but did not compare with it because it was not officially published at that time. Now, we add their results in our revised version. Please check our Tables 5 and 6, and you will find our method outperforms this recent baseline.
>
> Beyond, please check our Section 2.1, where we discuss and present some remarks on robustness theorems conditioned on symmetric losses and label noise. Our work challenges those robustness theorems, which can promote new thinking.
>
>
> 2. We explain our two reasons for without testing on asymmetric label noise:
> 1). As we mentioned in Section 4.2, we follow the prior work “Ma et al. Dimensionality-Driven Learning with Noisy Labels, ICML 2019” to test only on symmetric label noise as it has been demonstrated in Vahdat (2017) that it is more challenging than asymmetric noisy labels. Please check “Arash Vahdat. Toward robustness against label noise in training deep discriminative neural networks. NeurIPS, 2017.”
> 2). We spend more effort and space on experimental analysis and more complex and valuable real-world applications, e.g., image classification on Clothing 1M and video retrieval on MARS. $\mathit{Those \ problems \ are \ challenging \ and \ contain \ diverse \ semantic \ anomalies \ instead \ of \ only \ \ noisy \ labels.}$ For example, in Figure 3 in the supplementary material: $\mathit{1) \ Out-of-distribution \ anomalies}$: An image may contain only background or an object which does not belong to any training class; $\mathit{2) \ In-distribution \ anomalies}$: An image of class $a$ may be annotated to class $b$ or an image may contain more than one semantic object.
>
> $ \ \ $ Those experimental results prove that GR can achieve state-of-the-art performance on different domain tasks.
>
> Finally, we add the results of asymmetric label noise in Section $D$ of the supplementary material in our new revised version. The results are displayed in Table 10. When GR is used, the performance is better than its counterpart without GR.
>
> 3. The reference citation causes read difficulty.
> Thanks so much for pointing it out. We have put the citations into parentheses in the revised version.

---

### Decision · Program_Chairs · 2019-12-19

**Decision:**

Reject

**Comment:**

The paper proposes a gradient rescaling method to make deep neural network training more robust to label noise. The intuition of focusing more on easier examples is not particularly new, but empirical results are promising. On the weak side, no theoretical justification is provided, and the method introduces extra hyperparameters that need to be tuned. Finally, more discussions on recent SOTA methods (e.g., Lee et al. 2019) as well as further comprehensive evaluations on various cases, such as asymmetric label noise, semantic label noise, and open-set label noise, would be needed to justify and demonstrate the effectiveness of the proposed method.

---

> ### Author Response · Authors · 2019-12-26
> **Theoretical justification, Work of Lee et al. 2019, Code releasing**
>
>
> On theoretical justification, current theorems conditioned on symmetric loss values are not applicable and new theorems need to be built. Please see our viewpoints: https://openreview.net/forum?id=rylUOn4Yvr&noteId=BkfpxlP2or
>
> On recent SOTA methods (e.g., Lee et al. 2019), there is a difficulty, please see our provided information: https://openreview.net/forum?id=rylUOn4Yvr&noteId=BJerdkPhjB
>
> Many thanks to your helpful reviews. We will improve our work based on them. For your better reference, our key code is released now: https://github.com/XinshaoAmosWang/Emphasis-Regularisation-by-Gradient-Rescaling#1-code-is-available-now